# microRNA-122 amplifies hepatitis C virus translation by shaping the structure of the internal ribosomal entry site

Philipp Schult[1], Hanna Roth[1], Rebecca L. Adams[2], Caroline Mas[3], Lionel Imbert[3], Christian Orlik[1,5], Alessia Ruggieri[1], Anna M. Pyle[2,4] & Volker Lohmann [1]

The liver-specific microRNA-122 (miR-122) recognizes two conserved sites at the 5′ end of the hepatitis C virus (HCV) genome and contributes to stability, translation, and replication of the viral RNA. We show that stimulation of the HCV internal ribosome entry site (IRES) by miR-122 is essential for efficient viral replication. The mechanism relies on a dual function of the 5′ terminal sequence in the complementary positive (translation) and negative strand (replication), requiring different secondary structures. Predictions and experimental evidence argue for several alternative folds involving the miR-binding region (MBR) adjacent to the IRES and interfering with its function. Mutations in the MBR, designed to suppress these dysfunctional structures indeed stimulate translation independently of miR-122. Conversely, MBR mutants favoring alternative folds show impaired IRES activity. Our results therefore suggest that miR-122 binding assists the folding of a functional IRES in an RNA chaperone-like manner by suppressing energetically favorable alternative secondary structures.

[1] Department of Infectious Diseases, Molecular Virology, University of Heidelberg, Im Neuenheimer Feld 344, 69120 Heidelberg, Germany. [2] Department of Molecular, Cellular and Developmental Biology, Yale University, 219 Prospect St, New Haven, CT 06511, USA. [3] University Grenoble Alpes, CNRS, CEA, IBS, 71 Avenue des Martyrs, CS 10090, 38044 Grenoble CEDEX 9, France. [4] Howard Hughes Medical Institute, 219 Prospect St, New Haven, CT 06511, USA. [5] Present address: Department of Immunology, Molecular Immunology, University of Heidelberg, 69120 Heidelberg, Germany. Correspondence and requests for materials should be addressed to V.L. (email: Volker.Lohmann@med.uni-heidelberg.de)

The hepatitis C Virus (HCV) is a positive strand RNA virus. The genome consists of ~9600 nucleotides (nt) and contains a single open reading frame (ORF), flanked by 5′ and 3′ untranslated regions (UTR). The 5′ UTR contains a set of secondary structures that compose the internal ribosomal entry site (IRES), allowing a cap-independent translation of the viral genome. Cleavage of the resulting polyprotein by host and viral proteases gives rise to ten individual proteins, three structural proteins and seven non-structural (NS) proteins. The NS proteins are crucial for the formation of the membranous web, a vesicular structure containing the viral replication complexes. Genome amplification by the viral RNA polymerase initiates at the respective 3′ terminal structures of positive and negative strands[1].

The HCV 5′ UTR consists of four distinctly structured domains. Domain I (DI, nt 1–47, Fig. 1a) of the 5′ UTR is not essential for translation[2]. Except for nt 5–20, which fold into a small G:C rich hairpin structure (stem-loop I, SLI), the current model suggests an extended single-stranded form of the remaining sequence. In contrast, the complementary 3′ region of the negative strand is engaged in an extensive stem-loop (SLIIz′), which is crucial for RNA replication[3,4]. Therefore, both strands contain a structural code in this region, which is necessary for either translation or replication.

The HCV IRES harbors three distinctly structured domains (DII-IV), which serve as a binding hub for the eukaryotic translation machinery. DII can adopt an L-shaped tertiary structure, which is essential for IRES function as it mediates the displacement of eukaryotic initiation factor 2 (eIF2) complex to facilitate attachment of the 60S ribosomal subunit. Furthermore, it contacts the 40S ribosomal subunit. DIII is mostly responsible for host factor recruitment, it binds eIF2 and competes for the association of eIF3 with the 40S ribosomal subunit. DIV includes a short segment of the core coding region and folds into a pseudoknot element, which positions the viral AUG start codon in the ribosomal P site[5].

MicroRNAs (miRNA or miR) are 20–24nt long non-coding RNAs, which fine tune the expression of the majority of human genes[6] by inhibiting translation and accelerating mRNA decay. miRs are incorporated into an RNA-induced silencing complex (RISC), with one of four Argonaute (AGO 1–4) proteins as the central element. Target specificity is conferred by full complementarity of the seed region (5′ nucleotides 2–8), while the 3′ end is often imperfectly bound[7]. The 5′ end of the HCV viral genomic RNA features two conserved microRNA-122 (miR-122) target sites[8]. MiR-122 is liver-specific and highly abundant in hepatocytes, with over 60,000 copies per cell. It regulates lipogenesis, tumor suppression, and immune response[9]. The impact of miR-122 binding on HCV genome, however, is in stark contrast to the canonical mode of action of miRNAs. Upon binding of miR-122, HCV genome translation[10–12] and initiation of replication[13] are increased. Mir-122 binding further masks the HCV RNA 5′ end, conferring resistance to the degradation by host nucleases Xrn1 and Xrn2 and thereby contributing to increased protein production[14,15].

In this study, we show that stimulation of IRES activity substantially contributes to the diverse functions of miR-122 in the HCV replication cycle. Our in silico analysis predicted that the miR-122-binding region (MBR) has a negative effect on the formation of SLII, due to its additional *cis* functions in the complementary negative strand sequence. We show that mutations in the MBR favoring or disfavoring SLII formation indeed stimulate or abrogate translation, independent of miR-122. Our data therefore suggest that miR-122 binding suppresses alternative folds of the 5′ UTR that interfere with IRES function.

## Results

**Stimulation of HCV translation is critical for replication.** MiR-122 has been shown to be a vital host factor for HCV, as it enhances RNA stability, translation, and replication. We aimed to understand the contribution of translation enhancement to the miR-122 dependency of HCV, which is discussed controversially in literature, and to address the yet unknown mechanism.

To quantify the impact of miR-122 on HCV translation, we used full length (FL) viral genomes and subgenomic (SG) HCV replicons, lacking the viral structural protein coding regions, both harboring luciferase reporter genes (Luc) as integral parts of the viral polyprotein (Fig. 1a). We generated two variants of these constructs: One version followed the regular architecture of the HCV genome, in which translation of the polyprotein is initiated at the HCV IRES (Luc-SG, Luc-FL), causing RNA stability, replication, and translation to be dependent on miR-122. In an alternative version, we inserted a heterologous poliovirus (PV) IRES element, rendering translation of the HCV genome independent from miR-122 (Fig. 1a, PI-luc-SG, PI-Luc-FL). Huh7 hepatoma cells and subclones (e.g. Huh7.5 cells[16]) are the standard cell culture model for studying HCV replication cycle and contain high copy numbers of miR-122[9] (Fig. 1b). To allow the most thorough analysis of miR-122 effects, we used Hep3B hepatoma cells lacking detectable miR-122 expression (Fig. 1b). Reconstitution of miR-122 by co-transfection of a miR-122 duplex (miR^WT) rendered Hep3B cells permissive for HCV replication to similar levels as those measured in Huh7.5 cells[17] (Fig. 1b, d), whereas no replication was observed upon transfection of a miR-122 mutant with a point mutation in the seed region (Fig. 1c, d, miR^mut). To dissect the relative contribution of translation on miR-122 dependency of HCV, we co-transfected miR^WT or miR^mut with SG and FL replicons capable of miR-122 dependent and independent translation (Fig. 1a) into Hep3B cells (Fig. 1e–h) and determined the impact of miR-122 on translation. Specifically, translation of SG and FL replicons with the native HCV 5′UTR and HCV IRES driving translation of the polyprotein was affected most by exogenous miR-122, leading to an increase in luciferase activity by ~4–6-fold, compared with miR^mut (Fig. 1e, g, left panel). In contrast, those replicons with insertion of a heterologous miR-independent PV IRES showed less stimulation than miR-122-dependent reporters (Fig. 1f, h, left panels). This remaining increase in luciferase activity mediated by miR-122 was likely due to increased stability of the HCV genome, which was comparable in the range of 1.7-fold for luc SG and PI-luc-SG, since both constructs shared the same 5′ end (Supplementary Fig. 1a–c). Therefore, the net effect of miR-122 on translation efficiency of the HCV IRES was 2–3 fold, in line with previous reports[10–12]. Despite this apparently moderate additional impact of miR-122 on translation, the Luc-SG replicon did not replicate in absence of miR-122 (Fig. 1d, right panel), whereas insertion of the PV IRES allowed substantial replication levels under the same conditions (Fig. 1e, right panels). Similar results were obtained with full-length viral genomes (Fig. 1f, g). We validated these data in Huh7.5-based miR-122 knockout cells (Fig. 1b, d). Similarly, insertion of a miR-122 independent PV IRES element resulted in a significant increase in replication efficiency in the absence of miR-122 relative to a replication-deficient replicon (ΔGDD, Supplementary Fig. 1d–g). Due to the high reproducibility of data, we only used Hep3B cells in all subsequent experiments.

These results demonstrated that a moderate stimulation of HCV translation by miR-122 resulted in an overall substantial effect on viral RNA replication, indicating that translation significantly contributes to the miR-122 dependency of HCV.

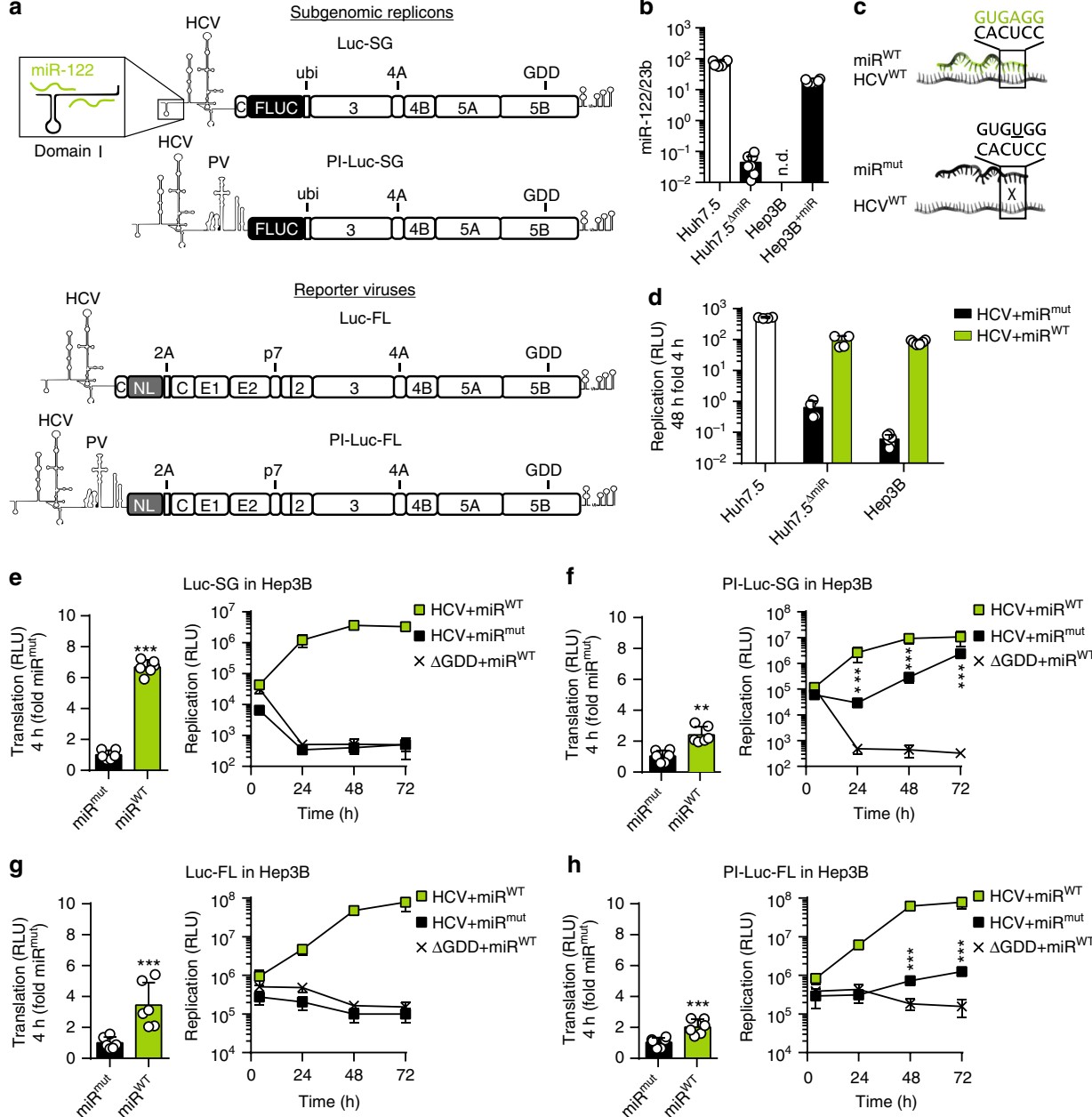

**Fig. 1** Initial translation and its importance for the HCV life cycle. **a** Representation of replicons and full-length viruses used in this study (FLUC firefly luciferase, NL nano luciferase). 5′UTR/IRES elements are indicated. The box shows an enlarged schematic of the miR-122-binding sites in the 5′ UTR. **b** Quantification of mature miR-122 by stem-loop qRT-PCR in naïve Huh7.5 cells, a Huh7.5 miR-122 knock-out cell clone (Huh7.5$^{\Delta miR}$), Hep3B cells before (Hep3B), and after (Hep3B$^{+miR}$) electroporation of miR-122 mimic. All values were normalized to miR-23b levels, which is homogenously expressed in hepatic cells. **c** Binding of wild type miR-122 (miR$^{WT}$) to a target RNA (HCV$^{WT}$) and influence of seed mutations (miR$^{mut}$). **d** Comparison of HCV replication in different cell lines by luciferase assay after co-electroporation of reporter replicons (Luc-SG) and miR$^{WT}$ or miR$^{mut}$ into Hep3B cells. Reporter activity was measured at 4 and 48 h. **e** Luciferase assay of initial translation (left panel) and replication (right panel) of subgenomic HCV. **f** Luciferase assay to determine the effects of PV IRES-driven polyprotein translation on the viral life cycle (PI-Luc-SG). **g, h** Confirmation of results from (**e**) and (**f**), using full length reporter viruses (Luc-FL, PI-Luc-FL). *n.d.* not detectable. Mean values (±SD), $n = 3$, in technical duplicates. *RLU* relative light units, ΔGDD replication deficient mutant. For translation, statistical significance was determined for miR$^{WT}$ against miR$^{mut}$; for replication HCV + miR$^{mut}$ was tested against ΔGDD. *$P<0.05$, **$P<0.01$, *$P<0.001$

**Stability and translation are essential for HCV replication**. To further disentangle the contribution of miR-122 on the processes of translation and/or stability/replication, a bicistronic luciferase reporter construct with two copies of the HCV 5′UTR was designed (Bi-Luc-SG, Fig. 2a). The first cistron consisted of HCV 5′ UTR (UTR1) harboring the cis-acting elements essential for RNA synthesis fused to a PV IRES allowing miR-122-

independent translation of the Nano luciferase (NLuc) gene. Translation of the second cistron comprising a *firefly* luciferase (FLuc) fused to the HCV NS proteins NS3-5B was initiated by the HCV IRES contained in the second copy of the HCV 5′UTR (UTR2). Since translation of the replicase proteins NS3-5B is a prerequisite for RNA replication, UTR1 and UTR2 thereby allowed studying the relative contribution of miR-122 to genome

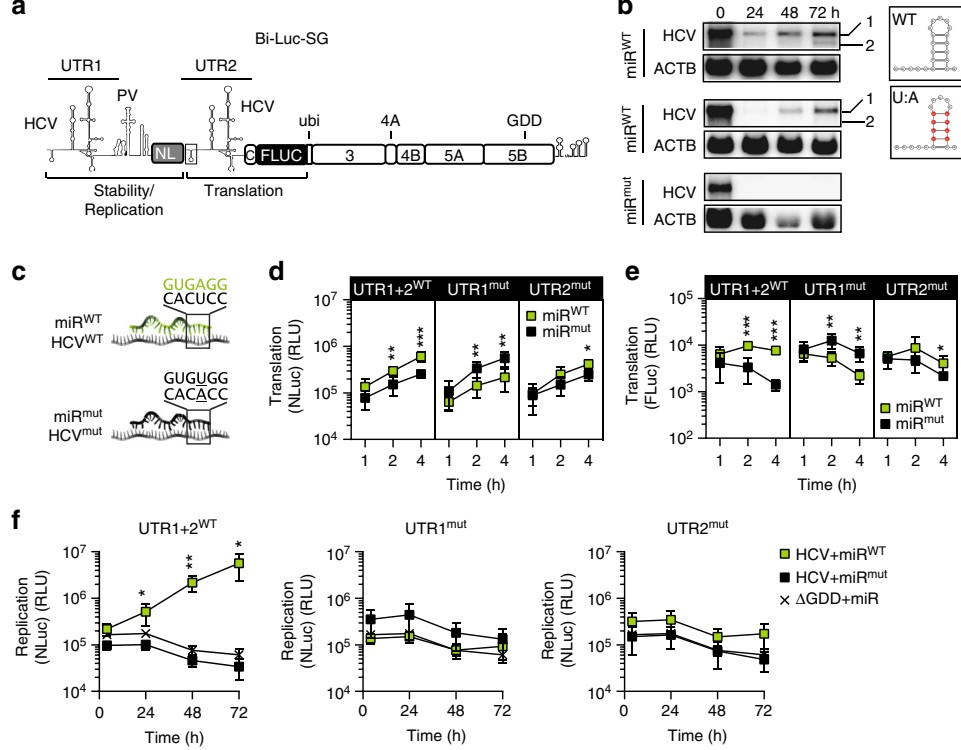

**Fig. 2** Dissection of stability and replication using a dual-luciferase replicon. **a** Illustration of the Bi-Luc-SG construct. miR-122 binding to UTR1 acts on RNA stability and replication (as measured by Nano luciferase activity), translation of non-structural proteins is driven by UTR2 (as measured by *firefly* luciferase activity). **b** Northern blot analysis of HCV RNA stability over 72 h. Stability of different Bi-Luc construct RNAs was analyzed in the presence of miR$^{mut}$ or miR$^{WT}$. Band #1: full-length RNA, #2: truncated product. U:A mutations in SLI of UTR2 (red dots) prevented the formation of truncated RNA. **c** Illustration of miR$^{WT}$ (green) and miR$^{mut}$ (black) binding upon introduction of a matching point mutation in the target (underlined). **d**, **e** Translation assay of point mutants cloned into the MBR of UTR1 or 2. The activity of Nano (**d**) and *firefly* (**e**) luciferase was measured in the presence of miR$^{mut}$ or miR$^{WT}$.
**f** Assessment of the full intracellular replication cycle of the Bi-Luc-SG constructs after electroporation with miR$^{mut}$ or miR$^{WT}$. Replication was monitored with the nano luciferase signal. Mean values (±SD), $n = 3$, duplicates. *RLU* relative light units, ΔGDD replication deficient mutant. For translation, statistical significance was determined for miR$^{WT}$ against miR$^{mut}$, for replication miR$^{WT}$ and miR$^{mut}$ conditions were tested against ΔGDD. *$P<0.05$, **$P<0.01$, ***$P<0.001$

stability/replication and translation, respectively, on genome replication.

Northern blot analysis revealed that two identical copies of the 5′UTR yielded truncated fragments at late stages of the replication cycle, likely due to sequence recombination (Fig. 2b, band #2). Therefore, the stem sequence in SLI of UTR2 was mutated from C:G to U:A, inactivating replication functions of UTR2[3]. This mutation prevented recombination without significantly affecting the replication efficiency of the bicistronic replicon (Fig. 2b).

To assess the individual contribution of miR-122 to HCV stability/replication and translation, we mutated UTR1 and UTR2 individually in the Bi-Luc-SG construct (Fig. 2a) by introducing a point mutation into the seed sequence of both miR-122-binding sites, complementary to the change in miR$^{mut}$ (Fig. 2c). Therefore, supplementation of miR$^{WT}$ or miR$^{mut}$ would only rescue miR-122-dependent functions in the respective matching UTR1 or 2 variants. Indeed, addition of either miR variant stimulated luciferase activities to varying extent early after transfection, either due to enhanced translation efficiency or RNA stability (Fig. 2d). For further control we deleted UTR1 and replaced UTR2 by an EMCV IRES to exclude crosstalk between UTR1 and UTR2 (Supplementary Fig. 2). These constructs revealed very similar results concerning miR-122 dependency as expected, apart from undetectable stimulation of translation by miR-122 upon deletion of UTR1, suggesting that in short-term experiments, the stabilizing effect of miR-122, preventing degradation

by Xrn1, seems to dominate in these constructs, probably due to their excessive length. However, none of the replicons harboring a mutant UTR sequence replicated to significant levels in the presence of miR$^{WT}$ or miR$^{mut}$ (Fig. 2e), demonstrating that both functions of miR-122 in stability and translation substantially contributed to efficient HCV replication. We further could rescue replication of a Bi-Luc-SG replicon harboring two mutant UTRs with miR$^{mut}$ (Supplementary Fig. 3a), as well as constructs with one mutant UTR by simultaneous addition of miR$^{WT}$ and miR$^{mut}$ (Supplementary Fig. 3b), albeit with very low efficiency. This was likely due to a generally lower efficiency of miR$^{mut}$ even in the context of a monocistronic replicon (Supplementary Fig. 3c) and a reduced stimulation by miR$^{WT}$ upon co-application of miR$^{mut}$ in case of UTR1 + 2$^{WT}$ Bi-Luc-SG (Supplementary Fig. 3d).

**SLII is affected by alternative conformations.** After establishing that miR122-driven translation stimulation is important to HCV replication, we investigated the underlying mechanism. Since the MBR itself is dispensable for IRES activity[2], the effect of miR-122 binding on the target sequence must be relayed from DI (nt1-47, Fig. 1a) to the downstream sequence to positively modulate translation. The current structural paradigm proposes a relaxed state of DI except for the stable SLI stem-loop. In contrast, the complementary region representing the 3′ end of the viral negative strand RNA (3′(-)) is engaged in a conserved stem-loop (SLIIz′, Fig. 3b), which is a *cis*-acting replication element (CRE)

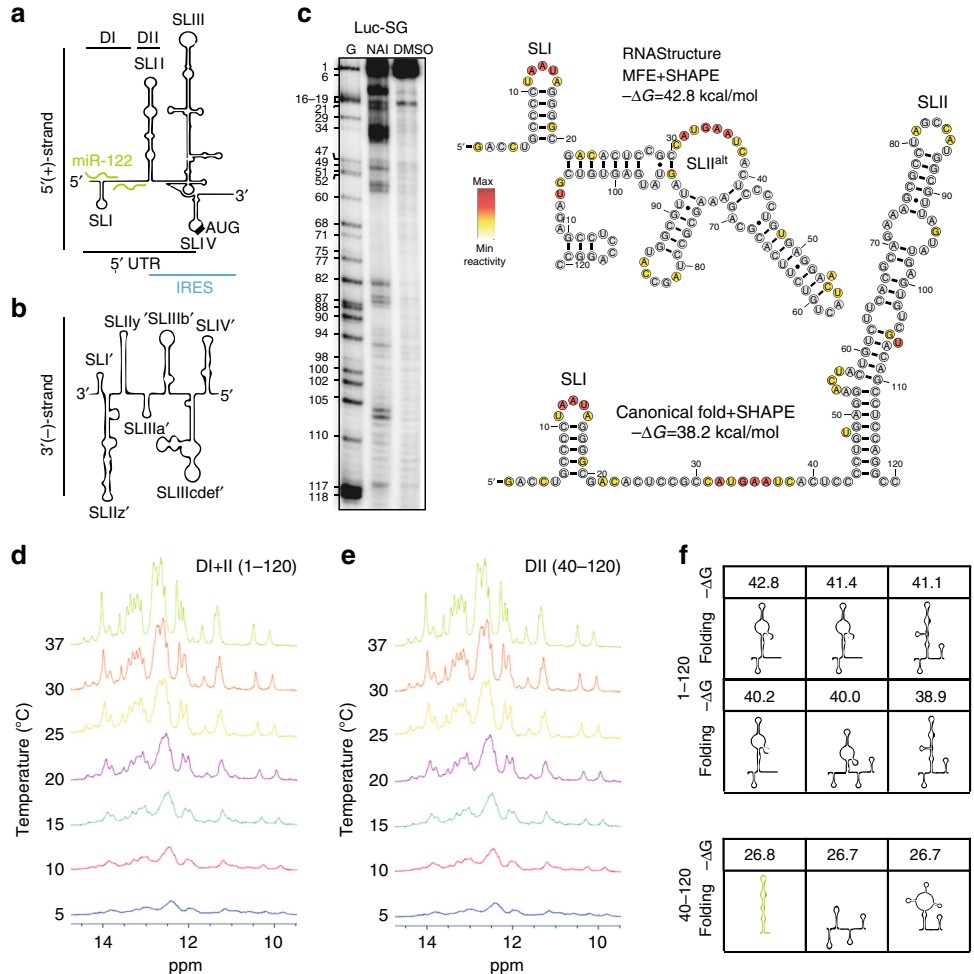

**Fig. 3** Structural assessment of the HCV 5′ UTR. **a** Schematic overview of the HCV 5′ UTR and IRES. The miR-122-binding sites are indicated in green. Domain I and II are indicated above (DI and DII). **b** Schematic overview of the 3′ UTR of the negative strand RNA, which is complementary to the 5′UTR of the positive strand. Note the divergence in secondary structures, which is due to the different function in replication (3′(-)-strand) and translation (5′(+)-strand). **c** Representative autoradiography image showing the in vitro SHAPE reactivity of nucleotides 1–120 of a HCV reporter replicon. The SHAPE data were used to predict the secondary structure (minimum free energy, MFE) of DI and II with the RNA structure web tool and were visualized with R2R. The nucleotides of SLII$^{alt}$ are colored according to their respective normalized reactivity. Yellow and red colors indicate unpaired nucleotides. The SHAPE data are also shown in context of the canonical fold of DI + II. **d**, **e** RNA conformation of DI + II (nt 1–120) or DII alone (nt 40–120), monitored by 1D $^1$H NMR spectra of imino protons in a temperature gradient. **f** Top conformations calculated by RNA structure and their respective free energy values ($-\Delta G$ [kcal/mol]) for DI + II or DII alone

crucial for RNA amplification[3]. Since complementarity of sequence also favors similar structures (except for U:G rich sequences), a structure similar to SLIIz′ might also form in the positive strand 5′ UTR, which would consequently prevent correct IRES formation. As the MBR is part of this alternative structure, miR-122 binding to this sequence could prevent misfolding of the IRES. Based on this hypothesis in vitro folding experiments of HCV RNA and in silico prediction of the conformation of DI and II were performed. Selective 2′-hydroxyl acylation analyzed by primer extension (SHAPE) reactivity data was used as constraint to assist structure prediction by RNA structure (Fig. 3c). In line with our hypothesis, the algorithm predicted an alternative stem-loop, encompassing nucleotides 21–105 as energetically most favorable option, from here on referred to as SLII$^{alt}$ (Fig. 3c), which partly resembled SLIIz′ in the negative strand RNA. Importantly, SLII$^{alt}$ also included parts of SLII, preventing the formation of a functional IRES. Alternative labeling by DMS and NAz produced both a pattern matching the prediction obtained with NAI (Supplementary Fig. 4a, b).

To gain deeper insights into the impact of DI on the folding of SLII, we subjected fragments of DI + II (nt 1–120) or DII (nt 40–120) to 1D-$^1$H NMR analysis. While the 1D NMR spectra of the DII imino protons were not significantly influenced by a decrease of experimental temperature (Fig. 3e), we observed line broadening of the imino proton resonances of the DI + II fragment at lower temperatures (Fig. 3d). This could indicate exchange between several conformers in solution at lower temperatures, or aggregates of RNA. To differentiate between these options, we performed size-exclusion chromatography combined with online detection by multi-angle laser light scattering (MALLS) and refractometry. The DI + II fragment was present as a monomer or dimer, both binding a copy of miR-122 not affecting their ratio (Supplementary Fig. 5a, b), whereas DII was exclusively monomeric and did not bind miR-122, as expected (Supplementary Fig. 5c). However, native gel electrophoresis at 4 °C with DI + II displayed only a single band, irrespective of the preincubation temperature (Supplementary Fig. 5d), further arguing against oligomerization. In addition, in silico folding analysis predicted six structures within the top 10%

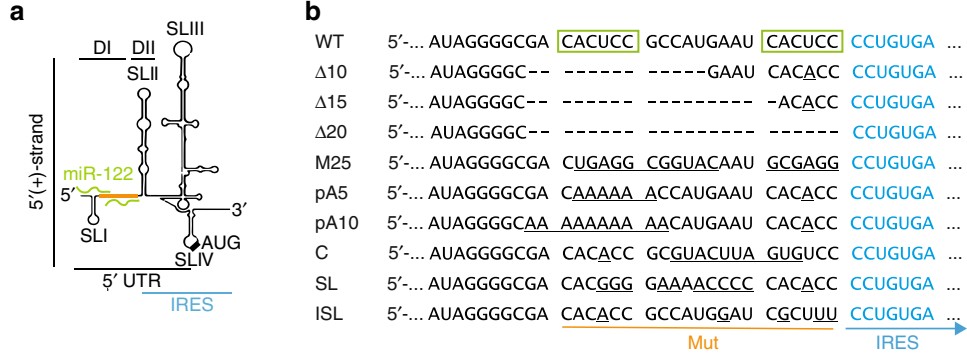

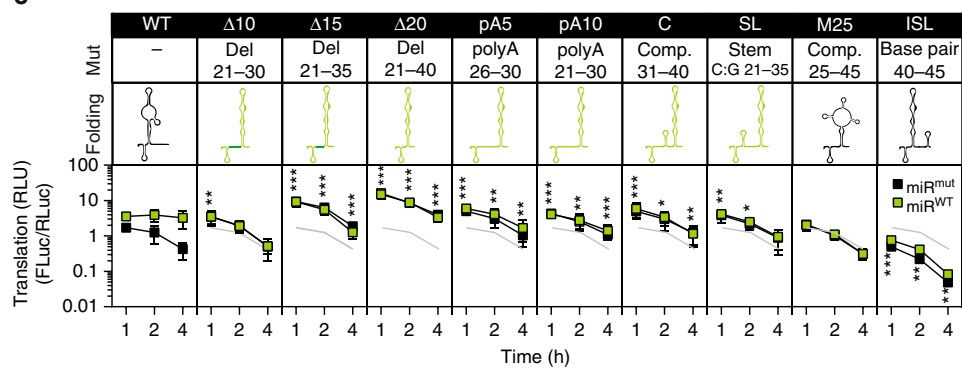

**Fig. 4** DI mutants and their influence on translation using the Luc-SG replicon. **a** Illustration of the HCV 5′ UTR and IRES and location of the inserted mutations (orange). **b** Sequence alterations of DI mutants compared to WT. The miR-122 seed binding sites are marked by green boxes. Note that all mutants lack wild type miR-122 binding, due to mutations in the seed sequence. **c** Translation assays in Hep3B cells, using *firefly* luciferase (FLuc) reporter replicons and a capped *Renilla* (RLuc) control. miR$^{mut}$ or miR$^{WT}$ were co-electroporated, and luciferase activity was measured after 1, 2 and 4 h. All constructs were replication deficient ΔGDD mutants, to exclude potential effects of early replication. Mutations are described in the "mut" row, and a pictogram of the predicted minimum free energy structure in the "folding" row. Mutants enhancing SLII formation are depicted in green, SLII$^{alt}$ stabilizing in red. Mean values (±SD), n = 3, duplicates. *RLU* relative light units. Statistical significance of the difference between WT and DI mutants in the presence of miR$^{mut}$ is indicated. The reference graph used to calculate is given in each subpanel in light gray. *P < 0.05, **P < 0.01, ***P < 0.001

of the minimal free energy structure for the DI + II fragment, in contrast to only three for the DII fragment (Fig. 3f), suggesting that the presence of DI increased the variety of alternative conformations of the 5′ terminus of the HCV RNA. Some of the energetically most favorable interfered with the formation of SLII.

We reasoned that binding of miR-122 would prevent the formation of alternative RNA structures in the positive strand RNA, and thereby support establishment of the IRES. To experimentally validate this assumption, we tried to assess the modulation of structures in the HCV 5′UTR by miR-122 using SHAPE. In vitro, miR-122 binding by annealing was efficient, indicated by an almost complete loss of SHAPE signal in the MBR, whereas, only moderate changes in reactivity were observed in the rest of the UTR (Supplementary Fig. 6a). Since in vitro binding of miR-122 might not reflect the situation in cellulo, we performed in cell SHAPE, which failed to detect any signal (Supplementary Fig. 4c). This result was in line with the previous data, identifying multiple secondary structures throughout the HCV genome, but not in the 5′UTR[18]. In addition, not many changes can be expected in the SHAPE signal, since most single stranded regions that are accessible by SHAPE in SLII$^{alt}$ and SLII are fully or partially overlapping (e.g. nt 53–60, 71–73, 80–86, 93–97, 105–108, Fig. 3b), with only a few exceptions (e.g. 63–64). Moreover, phylogenic analysis suggests that, due the high conservation of the primary sequence across all HCV genotypes, the sequence of the 5′ UTR is compatible with SLII$^{alt}$ folding. However, due to the few differences in primary sequence, which are also clustered within the bulge region of SLII$^{alt}$, we were not able to detect significant covariance to ascertain evolutionary

conservation of this structure (Supplementary Fig. 6b). However, several compatible mutations were detected in the corresponding complementary 3′ negative strand structure SLIIz′, rather arguing for selective pressure to keep this structure intact (Supplementary Fig. 6c).

In conclusion, our data suggest that DI, being essential for functions in the viral negative strand, interferes with the folding of the functional HCV IRES, thereby impairing genome translation, which is prevented by miR-122 binding. However, we were not able to directly detect the influence of miR-122 on downstream structures with the biochemical measures we implemented so far.

**Mutations in DI favoring SLII formation stimulate translation.** The previous results suggested a negative influence of DI on the conformation of the IRES, which can be prevented by miR-122 binding. To test this assumption, we designed a panel of DI mutants, all localized outside SLII and the IRES, which should either support or prevent the formation of SLII, according to secondary structure predictions (Fig. 4a, b). SLII formation was favored either by deleting parts of DI (Δ10, Δ15, Δ20), by replacing 5 or 10 positions with A (pA5, pA10) or by introducing mutations engaging DI into compensatory stem-loop structures (C, SL). As control, one mutant was designed, which was not able to fold stably into either SLII$^{alt}$ or SLII (M25), and another one was predicted to stabilize an SLIIz′-like fold (inhibitory stem-loop, ISL), preventing the formation of SLII. As expected, all mutants were severely impaired in replication or did not replicate

at all, most likely due to the destruction of essential parts of SLIIz′ in the negative strand (Supplementary Fig. 7a, Fig. 3b), therefore we focused our analysis on translation efficiency and stability (Fig. 4c, S3b–d).

Indeed, translation efficiency as measured by *firefly* luciferase activity, was strongly dependent on miR-122 for the WT UTR, whereas none of the mutants was influenced by miR-122, due to the lack of binding sites (Fig. 4b, c; Supplementary Fig. 7b). Still, all mutants designed to favor the SLII formation translated comparable to WT in the presence of miR-122 (Δ10, pA5, pA10, C, SL) or even substantially more (Δ15, Δ20) at early time points (Fig. 4c). In contrast, translation of the M25 was comparable to WT in the absence of miR-122 and could not be induced by miR-122, due to the absence of binding sites. Interestingly, the ISL mutant exhibited strongly decreased IRES-mediated translation far below the basal WT levels, in line with the predicted secondary structure, favoring SLII^alt formation (Fig. 4c).

Subsequently, we assessed the impact of the mutations on RNA stability and structure for a selected set of mutants and chose Δ20 (strongest stimulation), C (stimulation comparable to WT with miR^WT), and ISL (strong decrease in translation efficiency). We assessed short-term RNA decay kinetics by northern blot and found no correlation between stability and translation efficiency (Supplementary Fig. 7b–d). SHAPE analysis was performed to validate the successful generation of the desired structures by mutations in DI. Due to the limitations of SHAPE analysis discussed above, Δ20 and C did not show substantial differences compared to the WT. However, the ISL mutation, which was predicted to create a perfect mirror image of SLIIz′, indeed substantially changed the SHAPE pattern (Supplementary Fig. 8a–c).

To exclude inadvertent effects related to the changes in primary sequence of the MBR, we aimed to introduce mutations in SLII affecting SLII^alt and chose residues that are not in contact with the ribosome[19] (SLII^alt mut, C65G + G102C, Supplementary Fig. 9a). Indeed, SLII^alt destabilization lead to a miR-122 independent ~2-fold increase in early translation (Supplementary Fig. 9c, d). However, since these mutations were predicted to affect SLIIz′ folding (Supplementary Fig. 9e), they consequently abrogated replication (Supplementary Fig. 9f). Importantly, introducing compensatory mutations predicted to restore SLII^alt formation (SLII^alt rev, C49G, Supplementary Fig. 9b) again reduced translation efficiency (Supplementary Fig. 9c, d), but partially restored replication (Supplementary Fig. 9f).

Conclusively, mutations preventing or favoring alternative folds within DI and II substantially increased or decreased translation efficiency of the HCV IRES, independent from miR-122 binding. Our results demonstrate that the primary sequence of DI indeed negatively influences the activity of the IRES element by disturbing the fold of SLII, due to the essential role of the complementary sequence in RNA replication.

**miR-122-independent translation facilitates HCV replication.** To analyze how differences in translation efficiency exerted by DI mutations impacted overall replication efficiency, we used the Bi-Luc-SG construct (Fig. 5a). We compared a WT version of UTR1 and UTR2, keeping stability/replication and translation dependent on miR-122 with mutants harboring a UTR2 mutant either stimulating (Δ20, C) or reducing (ISL) translation efficiency independent from miR-122. NLuc activity at early time points after transfection was very similar and enhanced by miR-122 binding for all variants, reflecting increased RNA stability (Fig. 5b). FLuc activity was strongly stimulated by miR-122 binding in case of WT (Fig. 5c), whereas only a slight increase in FLuc acitivity was observed for the mutants due to the stabilizing effect of miR-122 upon binding to UTR1. Mutants Δ20 and C

showed FLuc levels comparable to WT in the presence and absence of miR-122, respectively, whereas FLuc levels of ISL were strongly reduced, in line with the previously demonstrated impact of these mutations on the folding of SLII. At later time points, no replication was observed for any construct in the absence of functional miR-122 (Fig. 5d, miR^mut), again demonstrating the important contribution of miR-122 to stability and replication of HCV genome. Yet, the mutants, all incapable of miR-122 binding, showed a divergent behavior: While the ISL mutant, impairing SLII formation, remained replication deficient even in the presence of miR-122, mutations Δ20 and C restored viral replication due to their capability to efficiently support SLII folding (Fig. 5c).

These data again emphasized the contribution of translation stimulation by miR-122 to HCV replication and provided further experimental evidence suggesting that mir-122 binding suppresses unfavorable folding of SLII.

**DI interferes with 80S ribosome assembly.** In the HCV IRES, SLII drives the assembly of the 48S pre-initiation complex with the 60S ribosome[20] (Fig. 6a). Hence, we hypothesized that introduction of mutations like Δ20, as well as miR-binding should enhance 80S monosome complex assembly, if these conditions favor SLII formation. Conversely, the ISL mutant should impair this transition.

To address, whether changes in DI had the propensity to modulate SLII function, 80S ribosome assembly was analyzed by polysome profiling. In vitro transcripts of Luc SG replicons (WT, D20, and ISL) were translated in vitro using miR-122-deficient HeLa cell extracts. Replicon translation efficiency was assessed by measuring *firefly* luciferase activity and showed comparable levels with those obtained in Hep3B cells (Fig. 6b). In vitro translation extracts were analyzed by sucrose density gradient centrifugation. Polysome profiles were continuously recorded and fractions collected based on the elution time (Supplementary Fig. 10a). Distributions of HCV RNA genomes were quantified in each fraction by qRT-PCR. Distribution of ribosomal RNAs was determined by agarose gel electrophoresis and allowed to distinguish fractions containing RNA associated with the 40S ribosome (18S rRNA) from those containing RNA associated with the 80S ribosome, i.e. engaged in translation. The fractions containing 40/48S and 80S ribosomes were further distinguished by adding 5′-Guanylyl imidodiphosphate (GMP-PNP) or cycloheximide (CHX), respectively (Fig. 6c). GMP-PNP prevents 60S recruitment by inhibition of eIF2 release, whereas CHX stalls the 80S complex via abrogation of the peptidyltransferase reaction[21]. Accordingly, the assembly of 80S ribosomes, indicated by the presence of 28S rRNA in higher density fractions, was impaired by GMP-PNP, but not CHX (Fig. 6c).

As predicted, addition of miR-122 to the WT replicon resulted in less HCV RNA present in the 40/48S fraction, concomitant with a higher amount in the 80S fraction, indicating a more efficient association with the 80S ribosome (Fig. 6d). In line with this observation, a 40S ribosomal subunit marker (rps3), was detected in higher density fractions after addition of miR-122, corroborating the results of the RNA gels (Supplementary Fig. 10b). The shift in 28S/18S rRNA, 40/48S:80S ratio and rps3 was even more pronounced for the Δ20 construct (Fig. 6e, Supplementary Fig 10b). In contrast, the ISL mutant was markedly impaired in its ability to initiate translation, since most viral RNA was retained at the 40/48S stage and only minute amounts of HCV RNA were detectable in higher density fractions (Fig. 6f). Expectedly, the distribution of eIF3, an essential translation initiation factor binding to DIII, was not altered in the mutants (Supplementary Fig. 10b). In contrast, rps3 was detected in the 80S ribosomal fraction (28S/18S rRNA,

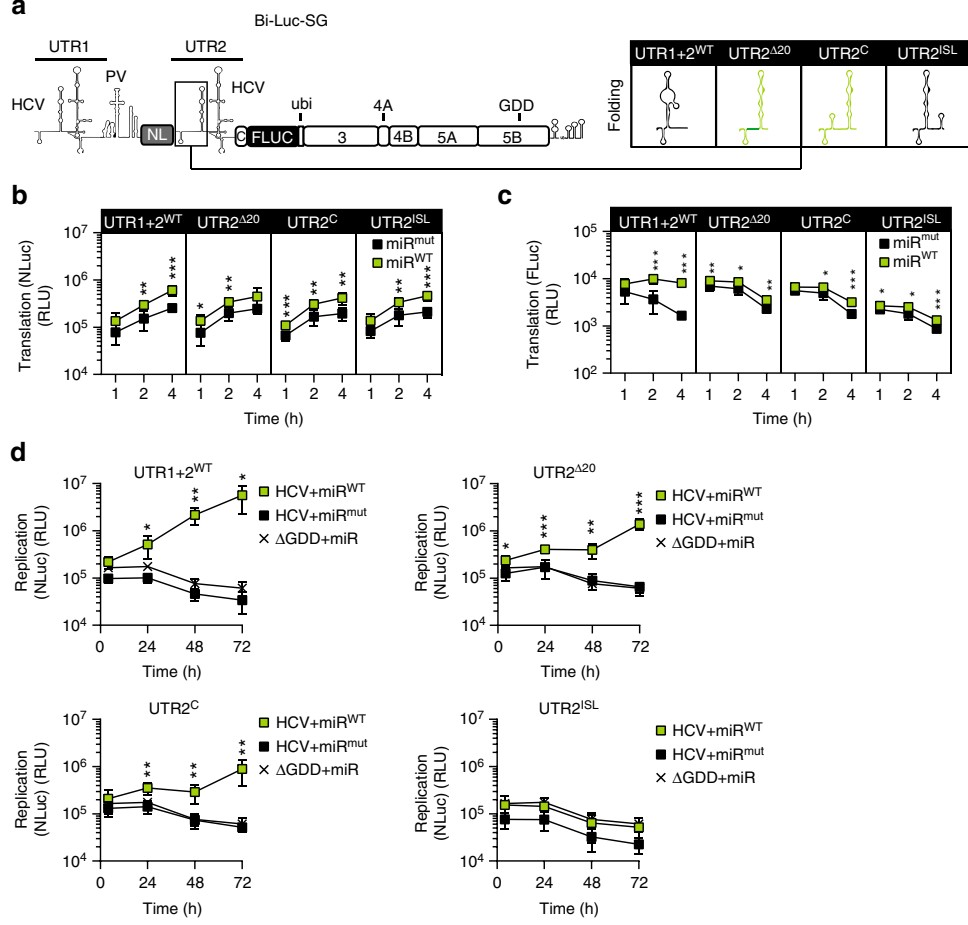

**Fig. 5** Translation stimulation by DI mutations in the Bi-Luc context. **a** Schematic representation of the dual-luciferase construct (Bi-Luc-SG) and insertion region of the mutations (black box). The in silico predicted MFE structures of the inserted mutants are given in the table on the right. Mutants enhancing SLII formation are depicted in green. **b**, **c** Translation assay of selected mutants cloned into UTR2. The activity of Nano and *firefly* luciferase was measured in presence of miR$^{mut}$ or miR$^{WT}$. **d** Assessment of the full intracellular replication cycle of the dual luciferase constructs after electroporation with miR$^{mut}$ or miR$^{WT}$. Replication was monitored via the Nano luciferase signal. Mean values (±SD), $n = 3$, in technical duplicates. *RLU* relative light units, ΔGDD replication-deficient mutant. *$P < 0.05$, **$P < 0.01$, ***$P < 0.001$

Supplementary Fig. 10b). These experiments were confirmed in presence of CHX, to stabilize transiently formed 80S subunits and prevent loss of signal due to polysome formation (Supplementary Fig. 10c). However, no significant differences were detected, perhaps due to the short translation time used in this experiment.

Overall, these results suggest that mir-122 binding stimulates the formation of SLII thereby favoring the assembly of the 80S ribosome and translation initiation. In addition, translation enhancement after miR-binding is mimicked by the SLII-forming mutant Δ20 but disabled in the ISL mutant.

**Natural DI variants stabilize SLII and increase translation.** Several HCV variants with reduced miR-122 dependency have been described but their mechanism of action remained elusive so far. The U3 mutant was obtained by a cellular recombination process, functionally replacing the SLI with a stem-loop from the U3 small nucleolar (sno) RNA (U3) and replicated in presence of miR-122 inhibitor and in Huh7.5 miR-122 knock-out cells[22,23]. Interestingly, in silico folding predicted native SLII formation of the U3 mutant in absence of miR-122 binding, which was reflected by strongly enhanced miR-122 independent translation (Supplementary Fig. 11a, b). Reintroduction of the missing miR-binding site was shown to restore miR-122 dependency (U3WTS1)[22]. In accordance with our hypothesis, mutant

U3WTS1 was predicted to preferentially form SLII$^{alt}$ (Supplementary Fig. 11a). This resulted in early translation efficiency and miR-122 dependency comparable to the WT UTR (Supplementary Fig. 11a, b).

Two independent studies selected for resistance to miR-122 antagonists or for miR-122 independence using miR-122 knock-out cells[24,25]. Both identified a single G to A point mutation in the 5′ UTR at position 28 (Fig. 7a) and demonstrated that this variant enabled HCV to replicate in non-hepatic and miR-122-deficient cells. We therefore assessed whether this mutant could enhance translation in the absence of miR-122. Indeed, a ~2-fold increase in luciferase activity was observed, compared to the WT, with a further increase upon addition of miR-122 (Fig. 7b). Accordingly, we confirmed a partial rescue of replication in the absence of miR-122 for the G28A mutant using the monocistronic Luc-SG replicon in Hep3B cells (miR$^{mut}$, Fig. 7c). RNA stability of mutant G28A was not significantly increased compared to the WT replicon (Supplementary Fig. 11c). To understand the mechanism underlying enhanced translation we performed in silico structure prediction of the mutant and WT (Fig. 7d, e). Strikingly, only the G28A mutant but not the WT sequence was predicted to form the functional SLII (green) within the top 10% of energetically favorable secondary structures (Fig. 7e), suggesting that the G28A mutation destabilizes deleterious structures and favors SLII formation. Therefore, the miR-independent

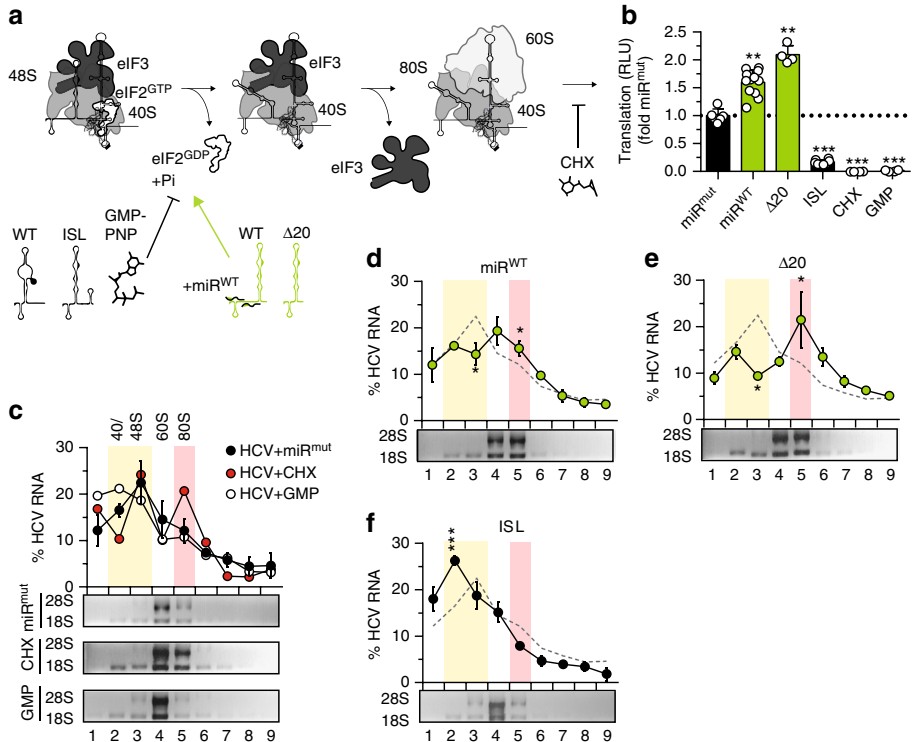

**Fig. 6** Ribosome profiling. **a** Schematic representation of translation initiation and assembly of the 80S ribosome. The 48S preinitiation complex is constituted of the 40S ribosomal subunit, eIF3 and the (eIF2)–GTP-tRNAP$^{Met}$P. Hydrolysis of GTP leads to release of eIF2. The L-shape of SLII is crucial for this process. Hence, the ISL mutant, misfolded WT and GMP-PNP, inhibiting GTPase activity, stall the reaction at this stage (bar-headed line). In contrast, miR-bound WT and Δ20 should activate eIF2-release, since SLII is effectively formed (green arrow). Subsequently, eIF3 is released and the 60S subunit is recruited to form the 80S complex. The following transition from initiation to elongation can be inhibited by CHX (bar-headed line). **b** Monitoring translation by luciferase assay in HeLa cell extracts, using Luc-SG in presence of miR$^{mut}$ or miR$^{WT}$, DI mutants, GMP-PNP or CHX. **c** Analysis of HCV RNA distribution in ribosome profiles of HeLa cell extracts. Total lysates incubated with HCV in vitro transcripts were recorded; Relative abundance of HCV RNA from sucrose fractions was analyzed by qRT-PCR. Represented are percentages relative to the total amount in the gradient. The assay was performed with either no inhibitor, GMP-PNP or CHX in the reaction mix. The HCV RNA profile for the control reactions is shown in the top panel. Total RNA was analyzed by denaturing agarose gel electrophoresis (lower panel) to confirm the presence of 18S and 28S ribosomal RNA and distinguish precisely 40/48S and 80S ribosomes. Note that in fraction 4 and 9 the CHX data points are obscured by the GMP data. **d** Comparison of translation initiation efficiency of Luc-SG in presence of miR$^{WT}$. The control with miR$^{mut}$ is shown as dashed line for comparison. **e, f** As in (**d**), comparing the WT reporter replicon and DI mutants. Mean values (±SD), $n = 3$, in technical duplicates. *RLU* relative light units. Significance was determined compared to the miR$^{mut}$ control. The reference graph used to calculate is given in light gray. *$P<0.05$, **$P<0.01$, ***$P<0.001$. 40/48S and 80S containing fractions are highlighted by yellow or orange boxes, respectively. Note that results shown in panels (**d**–**f**) were obtained in absence of CHX

replication capacity of the G28A variant is likely the result of enhanced initial translation due to an increased occurrence of SLII in the absence of miR-122.

In conclusion, both HCV UTR variants reported to replicate independent from miR-122 in cell culture (U3 and G28A), showed an increased translation efficiency in the absence of miR-122 and they are predicted to increase the formation of a functional SLII structure relative to the WT sequence. These data further corroborate our central hypothesis that miR-122 binding to the HCV 5′ UTR stimulates translation by suppressing unfavorable alternative secondary structure interfering with IRES activity.

## Discussion

In this study, we interrogated the mechanism by which the HCV IRES is activated in response to miR-122 binding. The process has been described extensively[10–12], but since the discovery of the RNA stabilizing effect[14,26,27], its relevance has been called into question. Our data now clearly demonstrate that translation stimulation by miR-122 can be functionally separated from RNA stabilizing effects by inserting heterologous, miR-122

independent IRES elements and by duplicating the 5′UTR to address stability and translation independently. Both approaches indicate that stability of the viral RNA of course indirectly impacts on translation, since protection from degradation will allow more sustained translation. However, we show that the direct stimulation of translation efficiency is an additional, independent effect of miR-122 in the range of 2–3-fold, in agreement with previous studies[10–12]. Importantly, this apparently minor effect substantially contributes to the miR-122 dependency of HCV. Our approach does not yet permit dissection of the stabilizing effects of miR-122 from additional RNA replication functions that have been proposed previously[13]. This can be addressed in future studies, comparing replication efficiency of the same series of constructs in cells lacking Xrn1 and Xrn2.

The sequence at 5′ end of the HCV genome serves two independent roles: IRES activity required for translation in the positive strand and essential replication signals in the complementary negative strand, both requiring very different secondary structures[3,4]. This dual function of complementary sequences prompted us to investigate a potential impact of miR-122 on IRES conformation. Interestingly, the thermodynamically most

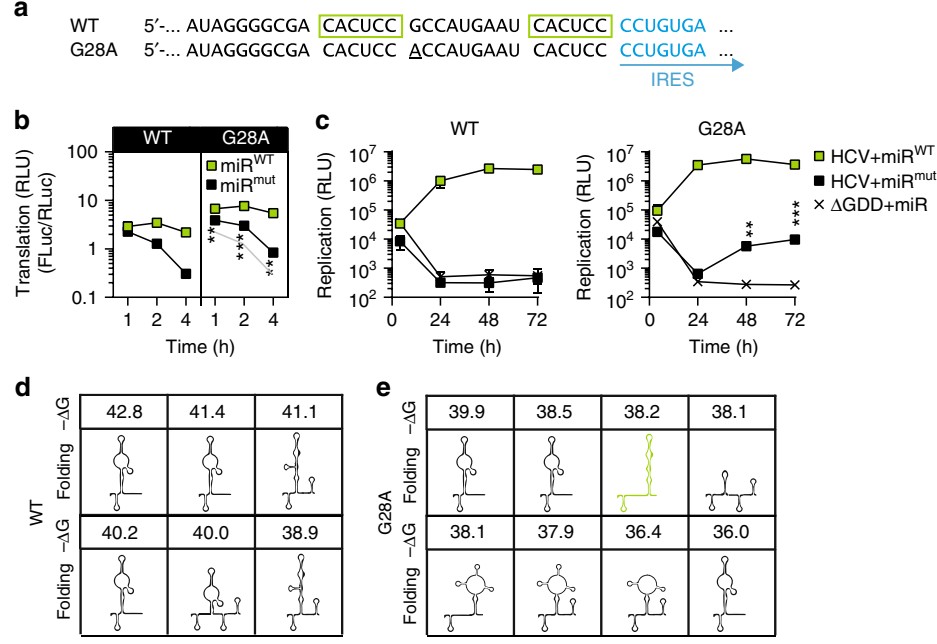

**Fig. 7** Impact of natural variants on IRES-mediated translation. **a** Sequence of the G28A mutant compared to wild type. **b** Translation assay of WT versus G28A in the presence of $miR^{mut}$ or $miR^{WT}$. **c** Replication assay of WT versus G28A in presence of $miR^{mut}$ or $miR^{WT}$. **d** Representation of the predicted structural ensemble for the first 120 nt of the WT, as predicted by RNA structure and their respective free energy values ($-\Delta G$ [kcal/mol]). **e** As in (**d**) for the G28A mutant. Mean values (±SD), $n = 3$, in technical duplicates. *RLU* relative light units, ΔGDD replication deficient mutant. For the translation assay, statistical significance of the difference between WT and G28A mutants in presence of $miR^{mut}$ is indicated. The reference graph used to calculate is given in light gray. $*P < 0.05$, $**P < 0.01$, $***P < 0.001$

favorable structures predicted by folding algorithms involved sequences of DI ($SLII^{alt}$), but not the active conformation of SLII required for IRES activity. Notably, a similar fold of this region was proposed earlier[28]. The structural diversity of DI and II was corroborated by NMR studies and in vitro SHAPE analyses in absence of miR-122 supported the inclusion of the MBR in a stem-loop-structure, rather than the canonical SLII fold. However, we were yet not able to directly demonstrate a structural change induced by miR-122. This was on one hand due the significant overlap in the predictions regarding single stranded regions between the different models. On the other hand, an active structural change might require the additional action of proteins, e.g. AGO, available only in cellulo. However, our attempts to show the presence of $SLII^{alt}$ by in cell SHAPE failed, which is in line with literature[18], suggesting that the 5′UTR of HCV is hardly accessible for unknown reasons. More elaborate in cell techniques will therefore be required in future studies to clarify whether miR-122 indeed prevents alternative RNA folds interfering with IRES activity.

We generated several mutants of DI predicted to either favor SLII formation or alternative structures interfering with IRES function. Alterations in DI that were designed to suppress deleterious structures, and in turn stabilize SLII, indeed activated early translation in absence of miR-122. Mechanistically, we could observe a shift towards 80S formation in ribosome profiling assays for the SLII stabilizing mutants Δ20 and C, comparable to the addition of miR-122 to WT genomes. These data argue for a stabilized functional SLII, as this feature promotes eIF2 dissociation, and enables 60S recruitment[20]. Accordingly, the ISL mutant, stabilizing $SLII^{alt}$ showed an inverse phenotype of abrogated translation and a block at the 40/48S stage. These results corroborate a previous report of Filbin and colleagues, observing a similar behavior of a ΔSLII construct[29]. The data obtained with the DI mutants therefore are the strongest support for our hypothesis of a strong impact of DI on the folding of a

functional SLII element. However, studies addressing HCV IRES structure often excluded DI, since it is not part of the IRES and thereby missed the negative impact on SLII formation[30].

Our data suggest that the predicted structural variation at the 5′ terminus of the HCV genome could simply be the result of the reciprocal evolutionary pressure on the active conformations of each strand (SLII vs. SLIIz′). Hence, miR-122 could assist the dual function of these complementary sequences and prevent deleterious folds. Currently it is not known whether miR-122 is already bound to the viral RNA within the virion or meets the 5′ UTR upon release of the genome into the cytoplasm of a newly infected cell. In the latter case, an incorrectly folded IRES will abrogate the infection process at an early stage. Even a 2–3-fold increase in the population of a functional IRES element, as suggested by the stimulating effect of miR-122 on translation, might have a tremendous effect on the efficiency of viral spread in vivo.

The closest relatives to HCV have been identified only recently in cattle, horses, bats, and rodents[31,32]. Interestingly, all examined specimens showed liver tropism and at least one conserved miR-122-binding site[32,33]. It is interesting to consider how such an intricate relationship between microRNA and virus has evolved. In fact, miR-122 and its liver specificity is conserved throughout the vertebrates. Therefore, the use of miR-122 to facilitate the formation of alternative RNA structures serving different functions might already have evolved in ancestors of recent *hepaciviruses* and it may have been maintained throughout co-evolution of virus and host, enforcing liver tropism. In case of GBV-B, a mutant lacking the 5′ terminus including the MBR is also attenuated, but viable[33].

Notably, a similar interplay between coding and non-coding RNA has been described in bacteria. Here, the ribosome-binding site of the RNA is occluded by an inhibitory structure, which is resolved by binding of an antisense RNA in complex with the bacterial AGO-analog Hfq[34]. Additionally, over the last decade there have been several reports of cellular IRES elements[35], which

were experimentally validated. Some of them also need chaperones to attain their functional conformation, expose the ribosome-binding site and activate translation[36]. Moreover, increasing numbers of functional miR-binding sites in the 5′ UTR of eukaryotic genes have been discovered, and some of these interactions can increase translation activity by an unknown mechanism[37]. These reports and our data suggest the possibility that sequence-specific RNA chaperones based on microRNAs may have originally evolved for regulatory requirements in eukaryotic cells.

## Methods

**Cells and cell culture.** Human Hep3B hepatoma cells (Prof. Ralf Bartenschlager, University of Heidelberg, Germany) were used as main model in this study. Huh7.5$^{ΔmiR}$ cells were generated by CRISPR/Cas9 technology using a lentiviral vector system and clonal selection. The founder cells Huh7.5[16] were a gift from Prof. Charles M. Rice (Rockefeller University, New York City, USA).

Human fibroblast HEK293T (Prof. Ralf Bartenschlager, University of Heidelberg, Germany) cells were used for generation of lentiviruses. All cell lines were grown in Dulbecco's modified Eagle medium (DMEM, Life Technologies) supplemented with 10% fetal calf serum (GE Healthcare/Sigma Aldrich), 100 μg/ml penicillin, 100 μg/ml streptomycin (Sigma Aldrich), and 1% non-essential amino acids (Life Technologies). Cells were maintained in a humidified incubator at 5% $CO_2$ and 37 °C. Huh7.5$^{ΔmiR}$ cells were cultured with 2 μg/ml puromycin (Sigma Aldrich).

**Generation of Huh7.5$^{ΔmiR}$ cells by lentiviral transduction.** Lentiviruses were produced in HEK293T cells by $CaPO_4$ transfection with pCMV-ΔR8.31 (HIV gag-pol), pMD2.G (VSV-G), and the pLenti-CRISPR-miR-122 vector in a 3:1:3 ratio, respectively. pCMV-ΔR8.31 and pMD2.G were kindly provided by Prof. Didier Trono (University of Lausanne, Switzerland). Cell-free supernatants were harvested 48, 56 and 72 h post transfection and used for transduction of Huh7.5 cells. Transduced cells were selected by supplementing the culture medium with 2 μg/ml puromycin. Clones were produced by seeding single cells in a 96-well plate, using a limiting dilution approach.

**Reporter viruses and plasmids.** FL HCV luciferase reporter viruses (Luc-FL, PI-Luc-FL) were based on JcR2a[38]. To achieve better signal at early timepoints, the *Renilla* luciferase was exchanged for a smaller and brighter Nano luciferase (Promega). pCMV-ΔR8.31 (HIV gag-pol) and pMD2.G (VSV-G) were used as packaging plasmids for lentivirus production. pLenti-CRISPR-miR-122 was produced by cloning a duplex of miR-122-specific sgRNA oligos (gRNA miR-122 sense oligo: CACCGAGTTTCCTTAGCAGAGCTG, gRNA miR-122 antisense oligo: AAACCAGCTCTGCTAAGGAAACTC), designed with E-CRISP (www.e-crisp.de) into the pLenti-CRISPR plasmid (Addgene #52961) via BsmBI (NEB). pFK-I$_{389}$Luc-ubi/NS3-3′/JFH1 (Luc-SG) was described before[39]. All mutants within the HCV 5′ UTR were generated by PCR-based mutagenesis and inserted with SbfI and XbaI (NEB). pFK-I$_{341}$PILuc-ubi/NS3-3′/JFH1 (PI-Luc-SG) has been used in a previous study[40]. It was generated by replacing the HCV IRES of pFK-I$_{389}$Luc-ubi/NS3-3′/JFH1 with the HCV 5′ UTR-Polio IRES cassette from pFK-I$_{341}$PILuc/NS3-3′/JFH1[41] using SbfI and XbaI (NEB). pRL-CMV (GenBank AF025843) was used for in vitro transcription of *Renilla* luciferase mRNA, as spike-in template for translation assays. pUC18 vectors were used for transcription of HCV 5′ UTR truncations (1–120nt, 40–120nt) for NMR analysis. The fragments were amplified from pFK-I$_{389}$Luc-ubi/NS3-3′/JFH1 plasmids with primers harboring SbfI and SmaI sites.

pFK-I$_{341}$PINLuc/I$_{389}$FLuc/NS3-3′/JFH1 (Bi-Luc-SG) was generated by insertion of a fusion PCR product of the I$_{341}$PI cassette from pFK-I$_{341}$PILuc/NS3-3′/JFH1 and the Nano Luciferase gene from pNL1.1.TK[Nluc/TK] Vector (Promega) into pFK-I$_{389}$Luc-ubi/NS3-3′/JFH1 before the HCV 5′ UTR. The HCV 5′ UTRs of the first and second cistrons were named UTR1 and UTR2, respectively. All mutants were generated by PCR-based mutagenesis and inserted with SbfI and AflII for UTR1, or AflII and XbaI (NEB) for UTR2. pFK-JcN2a (Luc-FL) was generated from pFK-JcR2a by amplifying the Nano luciferase gene from pNL1.1.TK[Nluc/TK] Vector (Promega) and adding the sequence of the 2A peptide. The NLUC-2A fragment was inserted into pFK-JcR2a with AscI and BsiWI (NEB).

pFK-JcPIN2A (PI-Luc-FL) was generated by replacing the I$_{389}$NLuc HCV IRES fragment from Luc-FL with the I$_{341}$-PI-NLuc cassette from pFK-I$_{341}$PINLuc/I$_{389}$FLuc/NS3-3′/JFH1 with SbfI and BsiWI (NEB).

**Oligonucleotides.** miR mimics were purchased at Eurofins MWG. Sequences of hsa-miR-122-5p (guide strand) and hsa-miR-122-3p (passenger strand) were acquired from miRbase (www.mirbase.org). However, an additional 3 U was added, since this sequence is the most abundant variant in hepatocytes[9] (hsa-miR-122-5p WT: UGGAGUGUGACAAUGGUGUUUGU, hsa-miR-122-3p WT: AAACGCCAUUAUCACACUAAAUA). For the inactive mutant miR-122$^{mut}$, A4 was changed to U, and the complementary exchange was performed on the passenger strand

(hsa-miR-122-5p mut: UGGAGUGUGACAAUGGUGUUUGU, hsa-miR-122-3p mut: AAACGCCAUUAUCACACUAAAUA).

**Stem-loop qRT-PCR for miR-quantification.** A previously described qPCR detection method for mature miRNA was adapted, using SYBR green instead of TaqMan probes for detection[42]. Total RNA was extracted from cells with TRIZOL (Thermo Fisher Scientific), according to the manufacturer's instructions. cDNA for miR-122-5p and miR-23b-5p was generated in a single reaction with specific RT primers (hsa-miR-122-5p RT primer: GTCGTATCCAGTGCAGGGTCCGAGGT ATTCGCACTGGATACGACCAAACA, hsa-miR-23b-5p RT primer: GTCGTAT CCAGTGCAGGGTCCGAGGTATTCGCACTGGATACGACGGTAAT). qPCR primers (50 nM, Stem-loop universal reverse qPCR primer: ATCCAGTGCAGG GTCCGAGG, hsa-miR-122-5p forward qPCR primer: GCGGCGGTGGAGTGTG ACAATG, hsa-miR-23b-5p forward qPCR primer: GCGGCGGATCACATTGCC AGGG) and dNTPs (0.5 mM) were preheated to 65 °C and flash-cooled on ice. 1× FirstStrand buffer (Thermo Fisher Scientific), 10 mM DTT, 1 U/μl RNAsin (Promega), and 2.5 U/μl SuperScript III (Thermo Fisher Scientific), and 0.5 μg of total RNA were added and samples were transferred into a TPersonal PCR cycler (Biometra) and cDNA synthesis was performed with a pulsed program: 30 min 16 ° C, cycle 60× (30 °C 30 s, 42 °C 30 s, 50 °C 1 s), 85 °C 5 min and then cooled to 4 °C. cDNA was diluted 1:20 and 3 μl were used for qPCR. Analysis was performed with the 2× iTaq Universal SYBR green supermix (Bio-Rad) on a CFX96 Touch real-time PCR detection system (Bio-Rad) with the following program: 95 °C for 3 min, 95 °C for 10 s, and 60 °C for 30 s. miR-23b was used as an internal reference, based on previous findings[43]. The 2$^{−CT}$ values of RT negative controls were subtracted for each sample. Relative gene expression was determined using the threshold cycle (2$^{ΔΔCT}$) method[44].

**In vitro transcription.** Sub-genomic reporter replicons: RNA for electroporation was generated from 5 μg of pFK-plasmid (linearized with MluI (NEB)) in a 100 μl reaction mix containing 20 μl of 5× rabbit reticulocyte lysate buffer (400 mM Hepes (pH 7.5), 60 mM $MgCl_2$, 10 mM spermidin, 200 mM DTT), 12.5 μl 25 mM NTP-solution, 2.5 μl RNasin (40 U/μl), 0.1 U pyrophosphatase (Sigma Aldrich), and 6 μl of T7 polymerase. The reactions were performed over night at 37 °C. Subsequently, the input DNA was removed using RQ1 RNase-free Dnase (Promega). The RNA was purified by acid phenol:chloroform extraction and isopropanol precipitation. The pellet was washed with 70% ethanol and resuspended in RNAse free water.

Capped *Renilla* mRNA for translation experiments: For capped *Renilla* transcripts, the pRL-CMV vector (Promega) was linearized with BamHI (NEB). The setup of the transcription reaction was as stated above, except 12.5 mM m7G analog (NEB) were added to the reaction, and the GTP concentration in the rNTP stock was reduced to 12.5 mM.

**Truncations of the 5′ UTR for NMR investigations.** 5′ UTR RNA constructs (1–120 nt (DI + II), 40–120 nt (DII)) for NMR studies were transcribed from pUC18 vectors, linearized using SmaI (NEB). 100 μg of linearized DNA plasmids were used for 1 ml run-off transcription reaction. The reaction mix containing 4 mM NTP, 16 mM $MgCl_2$, 34 μg of BSA, 3.6 U of pyrophosphatase from Becker Yeast, 125 μg of T7 RNA polymerase, 5 mM DTT, 1 mM spermidine, 0.01% Triton X-100, 80 mg of PEG 8000, and 40 mM Tris at pH 8, was incubated at 37 °C for 3 h in a Hybrigene HB-3D (Techne) at 20 rpm. Then, 50 units of RNAse-free DNAse (Euromedex) were added and incubated 30 min at 37 °C to stop the reaction. Reaction volumes were 1, 4, or 8 ml and RNA constructs were purified using Q-sepharose Hiload column (GE Healthcare) on DuoFlow (Biorad) in denaturing condition. Briefly, the reaction mixture was diluted 10-fold in 20 mM Tris buffer at pH 8, containing 10 mM EDTA and 8 M urea, and loaded on the column previously equilibrated with the same buffer. Then RNA was purified with a gradient from 0 to 1 M NaCl in the same buffer. Fractions containing RNA (identified using absorbance at 260 nm) were controlled using denaturing polyacrylamide (15%) gel electrophoresis. The fractions with the target transcript were pooled, dialyzed three times against 200 volumes of $H_2O$, and lyophilized.

The RNA samples for NMR studies were resuspended in water, heated at 95 °C for 5 min, and were rapidly cooled by the addition of ice-cold 2× NMR buffer. NMR buffer composition was 50 mM Hepes at pH 6.4, 100 mM NaCl, 5 mM $MgCl_2$ and 5% $D_2O$. The final RNA concentration for each NMR sample was 0.9 mM. All NMR experiments were recorded using a 950 MHz Bruker Avance III HD spectrometer equipped with a 5 mM cryogenically cooled pulsed-field-gradient triple-resonance probe.

**SEC–MALLS–RI.** Size exclusion chromatography (SEC) combined with online detection by MALLS and refractometry (refractive index—RI) is a method allowing the measurement of the absolute molecular mass of each component of a sample in solution, regardless of its dimensions and shape[45]. SEC was performed with a Superdex 200 column (GE Healthcare) equilibrated with 50 mM Hepes buffer (pH6.4) containing 100 mM NaCl and 5 mM $MgCl_2$. Fifty microliters of a DI + II and DII solution at 0.77 and 0.8 mM, respectively, was injected with or without 1.2 equivalents of miR-122. Separations were performed at room temperature with a flow rate of 0.5 ml/min. Online MALLS detection was performed with a DAWN-HELEOS II detector (Wyatt Technology Corp.) using a laser emitting at 690 nm,

and RNA concentration was measured online by measuring the differential RI using an Optilab T-rEX detector (Wyatt Technology Corp.) and an averaged RI increment, d$n$/d$c$, of 0.180 ml/g. Weight-averaged molar masses (Mw) were calculated using the ASTRA software (Wyatt Technology Corp.).

**Native gel analysis of RNA fragments.** 15 pmol of DI + II RNA fragment were denatured at 95 °C in 15 µl of $H_2O$ and flash cooled on ice. 8 µl of 3× SHAPE folding buffer (333 mM Hepes pH7.5, 333 mM NaCl, 20 mM MgCl$_2$) were added and reactions were incubated for 10 min at the desired temperature. Samples were put on ice to terminate the folding reaction and split into two tubes. One was supplemented with 3 µl of native loading dye (60 mM KCl, 10 mM Tris pH 7.6, 50% w/v glycerol, 0.01% w/v xylene cyanol, 0.01% w/v bromophenol blue), the other with 12 µl LDII (95% formamide, 0.025% SDS, 15 mM EDTA, 0.025% w/v xylene cyanol, 0.025% w/v bromophenol blue) as denaturing control.

10% polyacrylamide gels were pre-run for 10 min at running conditions. Native TBM (44 mM Tris–borate, pH 8.3, 1 mM MgCl$_2$) gels were run for 1 h at 100 V at 4 °C, denaturing urea TBE (89 mM Tris, 89 mM boric acid, 2 mM EDTA, 8 M urea) gels 1 h, 100 V at room temperature. Gels were stained with a 1:10,000 dilution of diamond nucleic acid dye (Promega) in the respective running buffer for 10 min at room temperature and visualized on a UV transilluminator (INTAS).

**Electroporation of cells.** 3*10$^6$ Hep3B or Huh7.5 cells were resuspended in a total volume of 200 µl in "Cytomix" (120 mM KCl, 0.15 mM CaCl$_2$, 10 mM K$_2$HPO$_4$/ KH$_2$PO$_4$ (pH 7.6), 25 mM Hepes (Gibco), 2 mM EGTA, 5 mM MgCl$_2$, 5 mM glutathione, 2 mM ATP) and electroporated (0.166 kV, 950 µF, 0.2 cm cuvettes) with 5 µg of reporter replicon RNA and 50 pmol wild type or mutant miR-122 mimic. For translation assays, 5 µg of a capped *Renilla* transcript were added as internal control for transfection efficiency. Electroporated cells were seeded onto six-well plates and incubated at 37 °C for the desired time.

**Luciferase reporter assay.** Activity of reporter replicons was measured after cell lysis in 350 µl luciferase lysis buffer (1% Triton X-100, 25 mM glycylglycin pH 7.8, 15 mM MgSO$_4$, 4 mM EGTA, 10% glycerol) per well in a six-well plate. Plates were stored at −20 °C until measurement of luciferase activity.

**Short-term translation assays.** Measurements of firefly and *Renilla* luciferase activity were performed separately in a Lumat LB 9507 single tube reader (Berthold).

For *firefly* luciferase, 360 µl luciferase assay buffer (15 mM K$_3$PO$_4$ pH 7.8, 25 mM glycylglycine pH 7.8, 15 mM MgSO$_4$, 4 mM EGTA) with freshly added 1 mM DTT, 2 mM ATP was mixed with 100 µl of lysate in a luminometer tube. After injection of 200 µl of 1 mM D-Luciferin (PJK) in 5 mM glycylglycine the sample was measured for 20 s. *Renilla* luciferase was measured by adding 100 µl luciferase assay buffer supplemented with 1.43 µM coelenterazine (PJK) to 20 µl of lysate, and an acquisition time of 10 s. Non-transfected cells were used to determine the background signal. Firefly luciferase from IRES-dependent translation, was normalized to the respective *Renilla* value to account for transfection efficiency and cell numbers. Note that capped in vitro transcripts encoding *Renilla* luciferase were not included in experiments primarily focusing on Northern blot analysis and in experiments using the Bi-Luc-SG replicon.

**Dual-luciferase assay.** For the dual luciferase replicons, the commercial Nano-Glo Dual-Luciferase Reporter Assay System (Promega) was used. In brief, 40 µl of cell lysate were transferred from the six-well plate to a white walled 96-well plate (Falcon) in duplicates. Next, 40 µl of ONE-Glo EX Reagent (Promega) were added, mixed and the reaction was incubated for 10 min at room temperature before measurement of the *firefly* luciferase signal. Then, *firefly* activity was quenched for 10 min using 40 µl of NanoDLR Stop& Glo buffer (Promega) before determining the Nano luciferase signal. Non-transfected cells were used to determine the background signal. Measurements were performed using a Mitras LB940 plate reader (Berthold).

**Total RNA extraction and Northern blot.** For each time point, 6*10$^6$ Hep3B cells were electroporated with 10 µg of replicon RNA. Cells from one electroporation were seeded on a 10 cm dish, and incubated at 37 °C for 0, 30, 60, and 120 min. Cells were scraped in the medium and put on ice immediately. 1 ml of the 6 ml suspension was transferred to an Eppendorf tube for luciferase assay, to control for translation activity at each time point. The cells were pelleted washed with PBS and taken up in 200 µl of Luciferase lysis buffer. The luciferase assay was performed as described above. The remaining sample was spun down and washed with ice-cold PBS with 0.1 mM aurintricarboxylic acid. Total RNA was extracted with acid guanidinium thiocyanate–phenol–chloroform and were denatured by treatment with 5.9% glyoxal in 50% DMSO and 10 mM NaPO$_4$ buffer, pH 7.0. Total RNA was separated by glyoxal agarose gel electrophoresis. Then, the RNA was vacuum blotted onto a nitrocellulose membrane and UV crosslinked twice in a Stratalinker 1800 (Stratagene) with the "Auto Crosslink" program. Blotting efficiency and RNA integrity was checked by methylene blue staining of the membrane. After washing the membrane with demineralized water, the blot was cut below the 28S rRNA

band and blocked with salmon sperm DNA at 68 °C in hybridization buffer. The upper slice was hybridized with a $^{32}$P-body-labeled RNA probe, complementary to nucleotides 5979–6699 of the HCV genome and the lower with a β-actin antisense riboprobe to correct for total RNA amounts loaded in each lane of the gel. Hybridization was performed over night at 68 °C. The blot was washed twice with wash buffer 1 (0.3 M NaCl and 30 mM trisodium citrate, 0.1% SDS) and three times with buffer 2 (0.03 M NaCl and 3 mM trisodium citrate, 0.1% SDS) at 68 °C. Specific bands were quantified after exposure of a phosphor imaging screen using a Molecular Imager FX scanner (Bio-Rad). Uncropped versions of the blots can be found in Supplementary Figs. 12–14.

**In silico RNA structure calculation.** For in silico folding of the 5′ terminal truncations of HCV, the online versions of RNA structure[46], mfold[47] were used without constraints. RNA structure was also used for integration of SHAPE data. Phylogeny analysis and visualization of RNA secondary structures was performed with Infernal[48] and R2R[49]. The sequences used are shown in Supplementary Data 1.

**Selective 2′-hydroxyl acylation analyzed by primer extension.** SHAPE is a method to address RNA structures in solution by chemical modification of flexible/ unpaired nucleotides. For readout, an RT reaction with a labeled primer is performed, and the polymerase will stop at each labeled residue. The resulting cDNA fragments are separated on a denaturing gel, and each band stands for single stranded RNA. Generation of 2-methylnicotinic acid imidazolide (NAI) and SHAPE assays were performed as describe before[50], with minor modifications. Briefly, 1 mmol 1,1′-carbonyldiimidazole (Sigma Aldrich) in 0.5 ml DMSO was added dropwise to an equal molar amount 2-methylnicotinic acid (Sigma Aldrich) in 0.5 ml DMSO, and the mixture was kept at room temperature for 1 h to generate NAI. In vitro transcripts or RNA isolated from cells were denatured for 2 min at 95 °C, flash-cooled on ice and refolded in presence or absence of miR-122-5p at 37 °C for 10 min in SHAPE buffer (100 mM HEPES pH 7.5, 100 mM NaCl, 6.6 mM MgCl$_2$). 100 mM NAI or DMSO were added and incubated for another 10 min. 1 µl β-mercaptoethanol was added to quench the reaction. 0.2 M NaOAc and 20 µg glycogen were added, and RNA was extracted with phenol:chloroform. RNA was precipitated with three volumes of absolute ethanol and washed twice with 70% ethanol. The purified RNA was dissolved in 12 µl RNAse-free water. For reverse transcription, 4 µl were denatured for 2 min at 95 °C with 1 pmol of a $^{32}$P-labeled primer (HCV-130: AGACCACTATGGCTCTCCCG, HCV-65: CTAGGCGCT TTCTGCGTG) and 0.5 M Betaine (Sigma Aldrich). After 2 min incubation on ice, 1× FirstStrand buffer (Thermo Fisher Scientific), 10 mM DTT, and 0.5 mM NTPs were added and the reaction was prewarmed to 55 °C for 1 min. 20 U SSIV reverse transcriptase (Thermo Fisher Scientific) were added and the reaction incubated for 10 min at 55 °C. Next, 1 µl of 4 M NaOH were added to hydrolyze the RNA and incubated at 95 °C for 5 min. The samples were cooled and one volume LDII was added before the cDNA fragments were separated by denaturing urea PAGE on a 10% gel (30 W, 180 min). The bands were quantified by phosphor imaging using a Molecular Imager FX scanner (Bio-Rad). SHAPE data were analyzed with SAFA[51]. The data were normalized to the mean of the top 10% reactivities. Intermediate reactivity (0.2–0.8) is indicated in yellow, highly reactive nucleotides (>0.8) are colored in red. For differential analysis of miR-bound versus free RNA the normalized values were subtracted and all differences of at least 0.3 were considered as significant. Less reactive nucleotides are depicted in green, more reactive in red.

For SHAPE, dimethyl sulfate (DMS) and nicotinoyl azide (NAz) probing of the full-length HCV genome, RNA was transcribed, purified, and folded as previously described[18]. Briefly, RNA was in vitro transcribed by runoff transcription with T7 RNA polymerase followed by treatment with RQ1 DNase (Promega). The RNA was then purified using RNeasy columns (QIAGEN) according to manufacturer's protocol and eluted in ME buffer (8 mM MOPS pH 6.5, 0.1 mM EDTA). The RNA was then adjusted to a final concentration of 250 ng/µL in folding buffer (50 mM K-HEPES (pH 7.4), 0.1 mM EDTA, 150 mM KCl, and 5 mM MgCl$_2$) and heated to 65 °C followed by cooling to 37 °C over a 45-min time course.

For DMS probing, DMS was added for a final concentration of 0.4%, or an equivalent volume of EtOH was added for unmodified control samples. Samples were incubated at room temperature for 10 min, and modification was quenched by addition of 2-mercaptoethanol at 10%. In case of NAz, the sample was exposed to UV light (310 nm) to activate the reagent.

Modified RNA was purified by phenol:chloroform extraction, LiCl precipitation, and resuspended in ME buffer at 1 µg/µl. Primers were $^{32}$P-end labeled by T4-PNK (NEB), and reverse transcription was performed with 1 µg RNA by SuperScript III (Thermo Fisher Scientific) according to manufacturer's protocol and the addition of 10U Superasin (Invitrogen). Following reverse transcription, RNA was degraded with the addition of NaOH and incubation at 95 °C, and cDNA products were resolved by denaturing UREA PAGE on an 8% gel.

**Translation assay in HeLa cell lysates.** 2.5 µg of reporter replicon RNA was added to 10 µl HeLa lysate (1-Step Human Coupled IVT Kit, Thermo Fisher Scientific) translation mix in a total volume of 15 µl, following the manufacturer's instructions. When needed, miR-122 duplex RNA was incubated at 30 °C in the mix, 10 min prior to RNA addition. The translation reaction was performed for 30 min at 30 °C. Then, 235 µl of ice-cold lysis buffer, containing 200 µg/µl CHX was

added to stop the reaction. 10 μl of the reaction were used as input sample for RNA extraction and western blot, 5 μl for luciferase assay. Tubes were kept on ice until fractionation. 5 mM GMP-PNP (Sigma Aldrich) or 500 μg/ml CHX (Sigma Aldrich) were added prior to the incubation for experimental determination of the 40/48S and 80S fraction, respectively.

**Fractionation on sucrose gradients.** Lysates were loaded onto a linear gradient of 17.5–30% in 15 mM Tris–HCl pH 7.4, 15 mM MgCl$_2$, 300 mM NaCl and subjected to ultracentrifugation at 165,000×$g$ at 4 °C using a SW60 rotor (Beckman) for 2.5 h. Gradients were then eluted from the top using a Teledyne ISCO gradient elution system. Polysome profiles were obtained by measuring the absorbance at 254 nm. In parallel, fractions were collected every 15 s (equivalent to ~400 μl). Before RNA extraction from the fractions, 20 μg of glycogen, 350 μl urea buffer (10 mM Tris–HCl pH 7.5, 350 mM NaCl, 10 mM EDTA, 1% SDS, 7 M urea) were added. Moreover, 1 ng of in vitro transcribed *Renilla* luciferase RNA was supplemented as internal control. Following phenol:chloroform extraction and ethanol precipitation, the pellets were resuspended in 40 μl of RNase-free water. Samples were stored at −80 °C until further processing.

**Detection of HCV RNA in fractions.** HCV genomic RNA extracted from the fractions was analyzed by one-step qRT-PCR using qScript XLT one-step RT-qPCR ToughMix (Quanta Bio-sciences) according to the manufacturer's instructions. Briefly, 15 μl of reaction mixture contained 7.5 μl of 2× RT mix, 1 μM of JFH1-specific and *Renilla*-specific primers (JFH1 S146 forward qPCR primer: TCTGCGGAACCGGTGAGTA, JFH1 A219 reverse qPCR primer: GGGCATAGA GTGGGTTTATCCA, *Renilla* S767 forward qPCR primer: AATCGGACCCAGGA TTCT, *Renilla* A917 reverse qPCR primer: ACTCGCTCAACAACGATTT), 0.27 μM of the corresponding probes (JFH1 TaqMan probe: [FAM]AAAGGACCCAGT CTTCCCGGCAATT[TAMRA], *Renilla* S859 TaqMan probe: [HEX]TCGCAAGA AGATGCACCTGATGA[TAMRA]), 3 μl of template RNA, and RNase-free water. To generate a standard curve, serial dilutions of an RNA standard ($10^7$–$10^9$ copies of HCV RNA and *Renilla* in vitro transcript) was prepared for each plate. Reactions were performed using the following program: 50 °C for 10 min, 95 °C for 1 min, and 40 cycles of amplification: 95 °C for 10 s and 60 °C for 1 min. The amount of HCV RNA was normalized to the *Renilla* control and the input sample.

**Immunoblotting.** 10 μl of each fraction were substituted with 2 μl of 6× Lämmli buffer with 5% β-mercaptoethanol and denatured for 10 min at 95 °C. Samples were separated on a 10% SDS polyacrylamide gel and wet-blotted on a PVDF membrane for 1 h at 400 mA. Blots were blocked with low-fat milk in PBS-T (0.1% Tween) for 30 min at room temperature. Primary antibody (goat polyclonal eIF3η antibody (Santa Cruz, N-20)), rabbit polyclonal RPS3 antibody (Abcam, ab140688) was applied in a 1:1000 dilution in milk PBS-T for 1 h at room temperature. Blots were washed three times with PBS-T and species specific HRP-coupled secondary antibodies were added in a 1:5000 (rabbit, Sigma Aldrich, A0545) or 1:10,000 (goat, Invitrogen, A15999) dilution. After another three washes, the blot was incubated with Clarity Western ECL Substrate (Promega) for 1–2 min and analyzed with an INTAS imager. Uncropped versions of the blots can be found in Supplementary Fig. 14.

**Denaturing RNA gel electrophoresis.** 18S and 28S ribosomal RNA was detected for each fraction using a denaturing TAE gel approach[52]. The samples were heated to 65 °C for 5 min in loading buffer LDII (95% formamide, 0.025% SDS, 15 mM EDTA, xylene cyanole 0.025% w/v, bromophenol blue 0.025% w/v) and flash cooled on ice. The denatured samples were applied to a 1% TAE agarose gel, containing 10 μg/L ethidium bromide. The gel was run for 30 min at 100 V in 1× TAE buffer and bands were visualized on a UV transilluminator (INTAS). Uncropped versions of the gels can be found in Supplementary Fig. 12.

**Quantification and statistical analysis.** Where appropriate, a paired, two-tailed Student's $t$-test was performed using Prism 6 software (GraphPad). All data are presented as mean (±SD), the sample size is specified in the corresponding figure legend. $P$ values of <0.05 were considered statistically significant. In graphs, statistical significance is indicated as follows: *$P < 0.05$; **$P < 0.01$; ***$P < 0.001$.

**Data availability.** The data that support the findings of this study are available from the corresponding author upon reasonable request.

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

## Acknowledgements

We are grateful to Rahel Klein and Ulrike Herian for excellent technical assistance. We thank Jérome Boisbouvier and Bernhard Brutscher (IBS, Grenoble, France) for aiding in interpretation of NMR data. The study was funded by a grant from the Baden-Württemberg Stiftung to V.L. (BWST_NCRNA_024). This work used the High Field NMR, Biophysical and Cell-Free facilities at the Grenoble Instruct-ERIC Center (ISBG; UMS 3518 CNRS-CEA-UJF-EMBL) with support from FRISBI (ANR-10-INSB-05-02) and GRAL (ANR-10-LABX-49-01) within the Grenoble Partnership for Structural Biology (PSB). C.M. benefited from an ANRS fellowship. A.R. was supported by the Deutsche Forschungsgemeinschaft (SFB 1129 TP13).

## Author contributions

Conception of the study and writing of the paper was done by P.S. and V.L. P.S., H.R., C. O., R.L.A., L.I. and C.M. performed experiments. H.R., C.O., R.L.A., L.I., A.M.P., A.R. provided valuable intellectual input and assisted in editing the manuscript. V.L. acquired funding.

## Additional information

**Competing interests:** The authors declare no competing interests.

