## [Peer Review File · Nature Communications]

Reviewers' comments:

Reviewer #1 (Remarks to the Author):

In this manuscript, Schult et al use a number of different approaches to determine that miR-122 regulates HCV translation by preventing formation of an inhibitory structure of the IRES. Overall, the data are convincing, and the approach of using the PV IRES to drive translation independent of the HCV IRES is elegant and informative. The observation that the PV IRES confers miR-122 independent replication on both subgenomic replicon and full length virus makes it clear that the effects of miR-122 on translation are important for its regulation of HCV. This paper makes an important contribution to the understanding of the mechanism by which miR-122 regulates HCV, resolving some of the current controversy on whether or not translation is involved.

Major comments

1. The translation data is plotted differently in different figures, with bar graphs at 4h shown in figure 1 and timecourses on a log scale shown in figures 2,4,5 and 7. While the timecourses are informative, it can be quite difficult to compare different mutants using these graphs. It would be useful to show a 4h bar graph as well, perhaps in supplementary data, to allow fold change in translation between different mutants to be clearly visualised.
2. In figure 4c and S7b, firefly luciferase translation is shown relative to a capped Renilla luciferase control, whereas translation data in the other figures is shown as raw values. This seems to result in some discrepancies in the data, in particular when translation profiles of mutants such as delta20 are compared in figure 4c and S3d, with differences also seen when the same mutations are made in UTR2 in figure 5b. The choice of this approach and possible explanations for the discrepancies need discussion. It is also not clear from the text or legend which constructs are used in figure 4, S3 and S7. Is this the Luc-SG replicon?
3. As the northern in figure 2b shows the appearance of truncated fragments of HCV, a larger portion of the blot should be shown to ensure that there are no further truncated fragments at a lower molecular weight.
4. To confirm that both functions of miR-122 in stability and translation are required for replication in figure 2e, the authors should also supplement with miR-122WT and mut in combination. This should restore replication of the mutants.
5. In figure 6, the authors should repeat the ribosome profiling of the mutants in the presence of CHX to obtain a clearer view of how 80S formation is affected. As they are shown, there is nothing to stabilize the transient 80S complex which may explain why the differences seen are

fairly small. Also, in figure 6c, data points appear to be missing for the CHX-treated samples in fractions 4 and 9. Is this because the red circles are obscured, or because of missing data? This should be clarified.

Minor comments

1. One of the references is given in full rather than as a superscript (line 84).
2. The authors should make it clear in the text and/or figure legends that data from figure 2 onwards were obtained in Hep3B cells.
3. The legend for figure 2d refers to top and bottom graphs, rather than left and right.
4. The sentence in line 144-146 is rather confusing. It should be rewritten to make it clear that the authors propose that a similar secondary structure to SLII-z' could form in the + strand, and that miR-122 binding to the MBR could prevent this.

Reviewer #2 (Remarks to the Author):

The study by Schult et al. examines effects of microRNA miR-122 on the translational efficiency of the internal ribosome entry site (IRES) in the hepatitis C viral (HCV) RNA genome. Binding of miR-122 has previously been shown to stabilize the viral RNA from 5' exonuclease attacks and to mediate a switch of the viral genome from translation to replication. A moderate, i.e. 2-4 fold, stimulating effect of miR-122 on IRES activity has been reported but its underlying mechanism is not known. Using elegant structural analysis, combined with genetic approaches, the authors show that binding of miR-122 upstream of the IRES favors the folding of domain II of the IRES to allow the formation of 80S ribosomes. Several chimeric and mutated viral genomes were used in this study to uncouple effects on translation from effects on replication and stability. The identified mechanism points to a model in which miR-122 plays a role in RNA folding that allows translation stimulation and end-to-end communication in the HCV genome.

Comments:

My main comment concerns the use of chimeric viral RNAs that contain three IRES elements in numerous experiments. How can the authors be sure that translation can be quantitatively measured at distinct IRESs? Clearly, each IRES will allow the binding of parts of the translation apparatus together with auxiliary factors that aid in polio- and HCV-IRES translation. There may be crosstalk between the IRESs. Have some of the experiments been carried out in non-replicating bicistronic constructs?

1. The interactions of SLI and SLIIz' should be shown already in Fig. 1

2. Fig S2a. A control miR should be used in this experiment

3. Fig. 6. Interestingly, HCV + mutmiR forms 48S complexes, but not 80S complexes. Does the expulsion of eIF2 require a miR-122-folded SLII complex? Is eIF2 enriched in HCV+mutmiR compared to HCV+CHX?

4. Fig. 6d-f. Do this samples contain CHX?

Reviewer #3 (Remarks to the Author):

Review of "microRNA-122 amplifies hepatitis C virus translation by shaping the structure of the internal ribosomal entry site" manuscript

This is a very interesting paper investigating the influence of HCV 5' UTR miR-122 binding sites on translation from the viral IRES. The authors demonstrate that miR-122 influence both the genome stability and the translation efficiency of HCV using co-transfection experiments of different HCV luc-replicons and a miR-122 duplex or a miR-122 mutant duplex. The use of the miR mutant as control is an elegant and relevant solution. The authors hypothesize that the effect on translation can be explained by a miR-122 mediated destabilization of an alternative IRES-SLII, which compete with the normal SLII required for internal ribosome recruitment. The alternative structure is based on in silico RNA folding guided by SHAPE data. The authors go on to show that the effect of miR-122 on translation can be influenced by mutations in the region designed to stabilize or destabilize an alternative IRES-SLII structure, which is not compatible with translation. The experiments included in the manuscript are well designed with good controls, the quality of the figures is high and the paper is very well written. The manuscript describes a novel mechanism potentially involved in the regulation of the HCV life cycle, which could also be of importance for researchers outside of the HCV field. Overall, I find that the manuscript is suitable for publication in Nature Communication, if the alternative SLII structure the effect of miR-122 on the structure can be further confirmed with additional probing experiments (see below).

A weakness of the paper, is the lack of data definitely showing that miR-122 binding affects the equilibrium between SLII and the alternative SLII. In fact, the in vitro probing data does not support structural rearrangement outside of the miR binding site. As discussed by the authors, the rearrangement could very well occur in vivo and be dependent of AGO protein. Moreover, the new mutants and the mutants identified in previous studies support the importance of the region for translation of the paper. In any case, a clear demonstration that SLII-alternative exists and that the miR dependant rearrangement take place would strengthen the conclusions in the paper.

This could potentially be demonstrated using in-native gel SHAPE probing, mutate-and-map or SHAPE-MaP. Likewise, if this the rearrangement happens only in vivo, in cell structure probing with NAI of HCV-replicons cotransfected with miR-122/miR-122mut/miR-control could further support the mechanism described in the paper.

Minor comments:

1. Figures in general. When printed the dark green coloring is difficult to distinguish from the black, at least for some color-deficient individuals.
2. Fig 1a: NL should be explained in legend or main text.
3. Figure 2d and 5b: The annotation of the y-axis is "Fold miR-mut", which seems to be a mistake. Looks like raw RLU.
4. Figure 4c: The figure shows significance between the WT construct and the Mutant construct, but plots the miR-WT and miR-mut values. This is quite confusing and should be changed.
5. Supplementary figure S2bc: from the figure and the main text, it is difficult to figure out if there is any evolutionary support for the alternative SL2, which is independent of the conserved structure on the negative strand. This would be interesting to know.
6. L377: Regulation of the translation/replication switch. This is an interesting possibility. Does the authors have any suggestions/speculation as to how this would be regulated.
7. L341: Rephrase: "Interestingly, the predictions did not rank the active conformation of SLII under the 10% thermodynamically most favorable structures"

Reviewer #4 (Remarks to the Author):

The HCV 5'UTR contains two binding sites for the liver-specific, highly abundant miR-122. The binding of miR-122 to the 5'UTR enhances stability of the HCV genome, its IRES-dependent translation as well as replication. The molecular mechanism by which all these functions are exerted remained unknown.

In this elegant study, the authors dissect the functions of miR-122 in viral translation and replication.

First they compared two wt HCV 5'UTR luciferase constructs where translation is either HCV IRES-dependent or IRES-independent in Hep3B cells lacking detectable miR-122 expression. Replication in the IRES-dependent construct was only observed upon co-transfection with miR-122 duplex but not in its absence or with a seed mutant of the miRNA. The HCV IRES-independent construct showed efficient replication in a miR-122 independent manner.

Translation, on the other hand, was only increased for the IRES-dependent construct in the presence of miR-122 duplex but not the seed mutant of the miRNA or the IRES-independent

construct. Subtracting the effects of the miRNA on the HCV RNA stability, the authors conclude a 2-3-fold increase in IRES-dependent translation mediated by miR-122.

This is further supported with a dicistronic constructs showing that miR-122 functions in stability and efficient translation are linked to efficient HCV replication.

The authors speculate that the miR-122 binding site in the HCV 5'UTR might affect the proper folding of the downstream domain II of the HCV IRES as well as the HCV CRE formed on the negative strand RNA during replication. Comparison of 1D NMR spectra comprising nt 1-120 of the 5'UTR (including the miR-122 binding site) or nt 40-120 (excluding the miR-122 binding site) show increased line broadening for the longer construct indicative of conformational exchange between alternative folds in the presence of domain I.

Comment:

While this is one possible explanation, it is also possible that the RNA simply oligomerizes by forming intermolecular weak base pairing between single-stranded nucleotides in domain I which are stabilized by lower temperatures. This could be easily studied by a simple gel filtration experiment comparing 1-120 and 40-120 constructs at different temperatures. In addition, miR-122 binding should be monitored by NMR. If miR-122 binding really promotes proper folding of domain II by binding the single-stranded region in domain I, the oligomerization should not be observed anymore with RNA 1-120 in the presence of miR-122 and characteristic imino protons of domain II should be clearly visible even at lower temperatures. In case of severe overlap, ¹⁵N labelling would be an option to monitor proper domain II folding induced by miR-122.

Comment:

In silico folding analysis suggests that the 1-120 construct forms a different domain II structure which is not translation competent. Unfortunately, SHAPE probing in the presence and absence of miR-122 could not detect significant changes in reactivity in the 1-120 HCV 5'UTR since single-stranded nucleotides are very similar in both predicted folds of the 1-120 construct. Using other probes such as DMS or CMCT might be more useful to confirm the alternative fold since the SLII-alt displays more cytosines, adenines as well as uracils in regions not forming canonical base pairs (see below).

Nevertheless, the data indicate that the first 40 nt of the 5'UTR are essential for replication but somehow hamper translation in the absence of miR-122. But “currently available biochemical measures” have not been exhausted to provide evidence for an alternative domain II fold.

The authors then design several deletion and point mutations in the miR-122 binding site that should favor proper domain II folding in the absence of miR-122 and thus promote translation. All these mutants fail to replicate but show translational efficiency comparable to wt (in the

presence of miR-122) unchanged by miR-122. A mutant lacking the miR-122 binding site and favoring an alternative fold in domain II, on the other hand, translated comparable to wt in the absence of miR-122 and could not be rescued by miR-122 addition. This provides genetic evidence that the proper, miR-122-dependent domain II folding is required for efficient HCV IRES-dependent translation.

P9, line 234:

miR-122 independent translational enhancement facilitates HCV replication.

Dicistronic vector: 1st wt UTR1 needs miR-122 for efficient translation, 2nd wt UTR2 also needs miR-122, but not the other mutants of UTR2 (D20, C, ISL). No replication was observed for any of the mutants without functional miR-122 but in the presence of miR-122 D20 and C mutants restored replication while ISL did not. I do not understand this or I am missing something?

D20 deletes the MBR so how does it work on the negative strand? Where is miR-122 binding in the SLIIz' fold of the negative strand? On p8, line 195, D20 was severely impaired in replication; but in the dicistronic vector all replication should occur via wt UTR1 in a miR-122-dependent manner anyway. So how is the difference in replication between D20, C and ISL explained?

P9, line 249:

What is the evidence that translation stimulation by miR-122 contributes to HCV replication?

For me these are independent processes, on the positive strand, miR-122 promotes and active IRES, on the negative strand, replication through interaction with a different RNA, where is the link?

Since domain II drives 60S subunit joining and 80S assembly, the authors perform sucrose density gradients after incubation of wt and mutant HCV 5'UTRs in HeLa cell extracts to monitor 48S and 80S formation. These experiments convincingly show that stabilization of domain II either by miR-122 or by promoting its proper fold (D20) leads to a shift towards 80S formation while destabilizing domain II (ISL) stalls at 48S stage.

P3, line 50:

Should be: which positions the viral AUG start codon in the ribosomal P site.

P9, line 250:

Should be: miR-122

Point by point response to reviewer comments

Reviewer #1 (Remarks to the Author):

In this manuscript, Schult et al use a number of different approaches to determine that miR-122 regulates HCV translation by preventing formation of an inhibitory structure of the IRES. Overall, the data are convincing, and the approach of using the PV IRES to drive translation independent of the HCV IRES is elegant and informative. The observation that the PV IRES confers miR-122 independent replication on both subgenomic replicon and full length virus makes it clear that the effects of miR-122 on translation are important for its regulation of HCV. This paper makes an important contribution to the understanding of the mechanism by which miR-122 regulates HCV, resolving some of the current controversy on whether or not translation is involved.

We are grateful to this reviewer for the very positive perception of our manuscript.

Major comments

1. The translation data is plotted differently in different figures, with bar graphs at 4h shown in figure 1 and timecourses on a log scale shown in figures 2,4,5 and 7. While the timecourses are informative, it can be quite difficult to compare different mutants using these graphs. It would be useful to show a 4h bar graph as well, perhaps in supplementary data, to allow fold change in translation between different mutants to be clearly visualised.

- We appreciate this suggestion. On the one hand we agree that the bar graphs as presented in fig. 1 are easier to follow, on the other hand we think that the time courses shown in the other figures are essential due to the varying and differential impact of stability and translation efficiency for the various mutants and conditions. In addition, the bar graphs showing translation in Fig1/Suppl. Fig. 1 are normalized data relative to control miR and deal with replication competent variants and thereby are complementing to the replication data, this was the reason to display them separately. In our view this way of normalization would not make sense for all mutants and conditions shown in the other figures, since many mutants cannot bind miR-122 anymore or do not replicate and the data are exclusively dealing with translation here. We are therefore reluctant to include fully repetitive datasets at so many points of the manuscript, since we think that this might be confusing for the reader. Still, we agree that particularly in Fig. 4 the number of mutants makes it very difficult to directly compare the data. To address this comment we therefore included an additional bar graph in Suppl. Fig. 7a, exclusively showing a single translation data point in comparison for all mutants.

2. In figure 4c and S7b, firefly luciferase translation is shown relative to a capped Renilla luciferase control, whereas translation data in the other figures is shown as raw values. This seems to result in some discrepancies in the data, in particular when translation profiles of mutants such as delta20 are compared in figure 4c and S3d, with differences also seen when the same mutations are made in UTR2 in figure 5b. The choice of this approach and possible explanations for the discrepancies need discussion. It is also not clear from the text or legend which constructs are used in figure 4, S3 and S7. Is this the Luc-SG replicon?

- We apologize for this apparent inconsistency, but the reason not to include Renilla transcripts in the experiment shown in Suppl. Fig. 3d is simply that the primary intention of this experiment was the Northern blot analysis, therefore also a different time scaling was used here, explaining the apparent discrepancy to Fig. 4; in fact the

data are almost identical. For the Bi-Luc-SG replicons we generally did not add another exogenous RNA, due the two luciferases already contained in the replicon. Here, the apparently different behavior of delta 20 (miR-122 dependency) in Fig. 5b is due to the presence of the wt-5' UTR, stabilizing the RNA.

To address this comment, we have explained these differences in the general experimental strategy in the Methods section. We furthermore added a sentence to the results section to explain the miR-122 dependency of the mutants in Fig. 5b. In addition, the type of replicons is now stated in the legends for Fig 4, S3 and S7 (now Suppl. Fig. 7 and 11).

3. As the northern in figure 2b shows the appearance of truncated fragments of HCV, a larger portion of the blot should be shown to ensure that there are no further truncated fragments at a lower molecular weight.

- We agree that this is important information. In fact, there are no smaller degradation products, and since Nature Communications requires the presentation of all uncut-blot images anyhow, which will be published along with the manuscript, the uncut blot displaying this information is now contained in a supplementary file for the readers' consideration.

4. To confirm that both functions of miR-122 in stability and translation are required for replication in figure 2e, the authors should also supplement with miR-122WT and mut in combination. This should restore replication of the mutants.

- We thank the reviewer for this constructive comment. Indeed, we expected as well that a combination of miR-122 WT and mut should rescue replication of the mutants. We did the suggested rescue experiments with either supplementing both miRs and the constructs containing the single site mutations, as well as only miR^{mut} with a double mutant replicon, which are presented in the new Suppl. Fig. 3. However, while we gained a partial rescue in all configurations, we were not able to fully restore replication levels in either configuration. A possible explanation for this is that adding both miRs results in a competition effect or overload of the cellular microRNA machinery. This might be deduced from the fact that even the wild type replicon was stimulated less efficiently by adding both miRs than after supplementation of just miR^{WT} alone (Suppl. Fig 3d). Moreover, we realized that miR^{mut} was not as efficient in its replication stimulating function as miR-122 WT, even in a monocistronic replicon Luc-SG (Suppl. Fig. 3c). Since this is a highly efficient replicon, the 2-5fold reduction was not appreciated by us at first. However, this effect will likely affect the less efficient BI-Luc-SG more strongly. In response to this comment we have added these new data to the manuscript (Suppl. Fig. 3) and discussed the results in the context of Fig. 2.

5. In figure 6, the authors should repeat the ribosome profiling of the mutants in the presence of CHX to obtain a clearer view of how 80S formation is affected. As they are shown, there is nothing to stabilize the transient 80S complex which may explain why the differences seen are fairly small.

- We have repeated this experiment in presence of CHX, as suggested by the reviewer and the new data are included in Suppl. Fig. 10c (red graphs). However, we could not see marked effects on 80S formation, likely due to the short translation time, not resulting in an accumulation of the 80S fraction even in presence of CHX.

Also, in figure 6c, data points appear to be missing for the CHX-treated samples in fractions 4 and 9. Is this because the red circles are obscured, or because of missing data? This should be clarified.

- We agree that this is confusing, but the apparently missing data points are indeed just obscured. We have now mentioned the overlap of data points in the respective figure legend.

Minor comments

1. One of the references is given in full rather than as a superscript (line 84).

- Sorry, this has been fixed.

2. The authors should make it clear in the text and/or figure legends that data from figure 2 onwards were obtained in Hep3B cells.

- Thanks for this hint; we agree that this is helpful for the reader. We have included a sentence in the results section to Fig. 1 specifying that Hep3B cells are exclusively used in the subsequent experiments of the manuscript.

3. The legend for figure 2d refers to top and bottom graphs, rather than left and right.

- Sorry, the figure legend has been corrected accordingly.

4. The sentence in line 144-146 is rather confusing. It should be rewritten to make it clear that the authors propose that a similar secondary structure to SLII-z' could form in the + strand, and that miR-122 binding to the MBR could prevent this.

- We have revised the sentence and added some additional explanations, as suggested by the reviewer, to avoid confusion at this point.

Reviewer #2 (Remarks to the Author)

The study by Schult et al. examines effects of microRNA miR-122 on the translational efficiency of the internal ribosome entry site (IRES) in the hepatitis C viral (HCV) RNA genome. Binding of miR-122 has previously been shown to stabilize the viral RNA from 5' exonuclease attacks and to mediate a switch of the viral genome from translation to replication. A moderate, i.e. 2-4 fold, stimulating effect of miR-122 on IRES activity has been reported but its underlying mechanism is not known. Using elegant structural analysis, combined with genetic approaches, the authors show that binding of miR-122 upstream of the IRES favors the folding of domain II of the IRES to allow the formation of 80S ribosomes. Several chimeric and mutated viral genomes were used in this study to uncouple effects on translation from effects on replication and stability. The identified mechanism points to a model in which miR-122 plays a role in RNA folding that allows translation stimulation and end-to-end communication in the HCV genome.

We thank this reviewer for the positive general comments on our manuscript.

Comments:

My main comment concerns the use of chimeric viral RNAs that contain three IRES elements in numerous experiments. How can the authors be sure that translation can be quantitatively measured at distinct IRESs? Clearly, each IRES will allow the binding of parts of the translation apparatus together with auxiliary factors that aid in polio- and HCV-IRES translation. There may be crosstalk between the IRESs. Have some of the experiments been carried out in non-replicating bicistronic constructs?

- Potential crosstalk of the different identical IRES elements is indeed a valid point that could impact on our results. We have therefore conducted short term translation experiments with constructs that lack the first HCV UTR, as suggested by the reviewer. In addition, we replaced the central HCV IRES by an EMCV IRES and also deleted HCV UTR1 in this context. Generally, the results of these experiments are well comparable to the full construct. The only inconsistency is that deletion of UTR1, thereby bringing the Poliovirus IRES towards the 5' end of the genome, renders the stimulating effect on miR-122 on translation from the HCV IRES undetectable. While this is unexpected, it reflects the tendency of UTR2 being less responsive to miR-122 than UTR1 in the Bi-Luc-SG context, which can also be seen in other experiments depicted in Fig. 2 and the new Suppl. Fig. 3. The Polio-IRES at the 5' end further generates unforeseeable effects on RNA stability, which might further impact on the measurement window for miR-122 effects on translation. This still does not affect the main conclusions that are based on these constructs, which are not dealing with the short term effects on translation, but the long term effects on replication. In response to this comment, we have added Suppl. Fig. 2 as well as text explaining this result to the results section of the revised manuscript. We further added Suppl. Fig. 3 in response to Reviewer 1, which further contains important information relevant for the overall interpretation of data based on the Bi-Luc-SG construct.

1. The interactions of SLI and SLIIz' should be shown already in Fig. 1

- We are not sure what the reviewer refers to here. We at no point of the manuscript indicate an interaction of SLI and SLIIz'. We think that the confusion might come from the arrow connecting both structures in Fig. 3a. To clarify this, we have removed this arrow and assigned an individual subpanel label to the 3'(-)-strand structure. In addition, we added explanatory text to the legend of Fig. 3 and the results section.

2. Fig S2a. A control miR should be used in this experiment

- We have repeated the SHAPE experiment using the miR-122-3p as control, as suggested by the reviewer. Since this did not show any differences to the condition without miR, apart from the miR-122 binding site, we decided not to add an additional Supplementary Figure, but to present the result as Reviewer Figure 1 here.

Reviewer Figure 1: Autoradiography image of in vitro SHAPE in presence of single stranded miR-122-5p or miR-122-3p control, capable or not capable of binding to the HCV 5'UTR, respectively, using NAI reagent. Note that except the different reactivity in the miR-122 binding region (see also prev. Suppl. Fig. 2a, now 6a), no differences are observed.

3. Fig. 6. Interestingly, HCV + mutmiR forms 48S complexes, but not 80S complexes. Does the expulsion of eIF2 require a miR-122-folded SLII complex? Is eIF2 enriched in HCV+mutmiR compared to HCV+CHX?

- According to current literature, eIF2 and eIF3 release is dependent on a functional SLII. We followed the reviewers suggestion and analyzed the distribution of eIF2a by Western blot in the gradient fractions as well. However, eIF2 signal was barely detectable and could only be found in the total lysate (Reviewer Fig. 2). Due to this overall negative outcome, which does not allow further conclusions, we did not mention this result in the revised manuscript.

Reviewer Figure 2: Western blots against eIF3 η and rps3 as markers for pre 80S complexes and small ribosomal subunit, respectively, corresponding to the fractions shown. Total lysate is shown on the left. Note that eIF2a could not be detected in the gradient fractions and is only visible as a very faint band in the total lysate on the long term exposure on the right (blue arrowhead).

4. Fig. 6d-f. Do this samples contain CHX?

- The samples shown in Fig. 6d-f did not contain CHX. We have repeated the fractionation with added CHX, according to this suggestion and the request of Reviewer 1. The data is presented in Suppl. Fig. 10c (red lines). We further mention the absence of CHX in the legend of Fig. 6.

Reviewer #3 (Remarks to the Author):

Review of "microRNA-122 amplifies hepatitis C virus translation by shaping the structure of the internal ribosomal entry site" manuscript

This is a very interesting paper investigating the influence of HCV 5' UTR miR-122 binding sites on translation from the viral IRES. The authors demonstrate that miR-122 influence both the genome stability and the translation efficiency of HCV using co-transfection experiments of different HCV luc-replicons and a miR-122 duplex or a miR-122 mutant duplex. The use of the miR mutant as control is an elegant and relevant solution. The authors hypothesize that the effect on translation can be explained by a miR-122 mediated destabilization of an alternative IRES-SLII, which compete with the normal SLII required for internal ribosome recruitment. The alternative structure is based on in silico RNA folding guided by SHAPE data. The authors go on to show that the effect of miR-122 on translation can be influenced by mutations in the region designed to stabilize or destabilize an alternative IRES-SLII structure, which is not compatible with translation. The experiments included in the manuscript are well designed with good controls, the quality of the figures is high and the paper is very well written. The manuscript describes a novel mechanism potentially involved in the regulation of the HCV life cycle, which could also be of importance for researchers outside of the HCV field. Overall, I find that the manuscript is suitable for publication in Nature Communication, if the alternative SLII structure the effect of miR-122 on the structure can be further confirmed with additional probing experiments (see below).

- We thank this reviewer for the positive general comments and the very constructive suggestions to improve our study. We tried hard to demonstrate direct effects of miR-122 on RNA structure, following these suggestions, as well as the suggestions of Reviewer 4. We have performed experiments with NAI, DMS and NAz. All reagents show a band pattern in agreement with the existence of SLIIalt, arguing for the initial hypothesis of the existence of an alternative structure in solution. Unfortunately, despite many independent attempts in two different labs, we were not able to generate in cell SHAPE data. Due to the fact that also in literature no such data have been presented so far, this might reflect a more general problem with the very 5' end of the genome, since in cell SHAPE was successful in other regions of the HCV genome as demonstrated by our co-author Anna Pyle (Pirakitikulr et al., Mol Cell 2016, PMID: 26924328).

A weakness of the paper, is the lack of data definitely showing that miR-122 binding affects the equilibrium between SLII and the alternative SLII. In fact, the in vitro probing data does not support structural rearrangement outside of the miR binding site. As discussed by the authors, the rearrangement could very well occur in vivo and be dependent of AGO protein. Moreover, the new mutants and the mutants identified in previous studies support the importance of the region for translation of the paper. In any case, a clear demonstration that SLII-alternative exists and that the miR dependant rearrangement take place would strengthen the conclusions in the paper. This could potentially be demonstrated using in-native gel SHAPE probing, mutate-and-map or SHAPE-MaP. Likewise, if this the rearrangement happens only in vivo, in cell structure probing with NAI of HCV-replicons cotransfected with miR-122/miR-122mut/miR-control could further support the mechanism described in the paper.

- We agree that especially *in vivo* data could significantly strengthen the hypothesis. Therefore, we have tried in cell SHAPE (based on the approach of Spitale et al. 2013) after electroporation to detect the phase of initial translation. Unfortunately, we were not able to obtain any signal, although the positive control 5S RNA did work. Also, library generation for SHAPE-MaP and the detection of signal for the HCV 5'UTR in replicating cells did not yield any result, so far (data not shown). It further has to be considered that the alternative structures will be rapidly degraded in living cells by Xrn1 since no miR-122 is bound. In addition, in native gels the 120nt fragment did not show multiple bands, therefore native gel SHAPE was not possible. We will try mutate-and-map in the future, but this was not feasible within the time frame of the revision. Still, one overall limitation of all approaches is the relative moderate difference between the two proposed structures and therefore also limited options to visualize any impact of miR-122 binding. Still, we think we can provide overall convincing evidence for the existence of alternative structures in absence of miR-122, now supported by two additional reagents (DMS and NAz, Suppl. Fig. 4). In response to this comment we have further included the negative data on in cell SHAPE including the positive control (Suppl. Fig. 4c). We further show the lack of alternative bands in native gels (Suppl. Fig. 5d) and discuss the limitations of our results extensively in the revised manuscript.

Minor comments:

1. Figures in general. When printed the dark green coloring is difficult to distinguish from the black, at least for some color-deficient individuals.

- We apologize for this difficult to differentiate colors. We have lightened up all green color throughout the manuscript to allow a better differentiation from the black.

2. Fig1a: NL should be explained in legend or main text.

- The abbreviation is now explained in the respective figure legend.

3. Figure 2d and 5b: The annotation of the y-axis is "Fold miR-mut", which seems to be a mistake. Looks like raw RLU.

- Indeed the reviewer is right, this has been corrected.

4. Figure 4c: The figure shows significance between the WT construct and the Mutant construct but plots the miR-WT and miR-mut values. This is quite confusing and should be changed.

- We agree that this presentation is confusing. To make the basis of the significance testing clearer to the reader, we have copied the reference graph (Wt construct with miRmut) into every subpanel in light gray and explain this in the legend to the figure in greater detail. In addition we have added a selected view on the 1 hour values as a

bar graph in Suppl. Fig. 7a, including the significance tests, this should further help to make this point clearer.

5. Supplementary figure S2bc: from the figure and the main text, it is difficult to figure out if there is any evolutionary support for the alternative SL2, which is independent of the conserved structure on the negative strand. This would be interesting to know.

- This is indeed an interesting point. However, in our view it is rather unlikely that the alternative SL has a function in the positive strand, but that the selective pressure comes from the well documented functions in the negative strand. Generally, the sequence conservation in this region is too high to allow firm conclusions. We have now added a sentence stating that, due to the high sequence identity in this region, we cannot claim evolutionary conservation of SLII^{alt} based on this analysis.

6. L377: Regulation of the translation/replication switch. This is an interesting possibility. Does the authors have any suggestions/speculation as to how this would be regulated.

- As mentioned above, we regard this option as a theoretical option, based on a similar structure in the flavivirus genome, however, flaviviruses do not contain an IRES. In the light of miR-122 such a mechanism seems rather unlikely, since translation will require miR-122 binding, as shown by our study and by others before and it is hard to figure out, how miR-122 should be removed to allow a switch to replication. Due to a substantial increase in word count of the results section to respond to the reviewers suggestions, we needed to cut other sections of the manuscript to stay within the limits allowed by the journal. Since this part of the discussion was in our view highly speculative and since we have no clear vision how this should be accomplished, we rather removed this section of the discussion.

7. L341: Rephrase: "Interestingly, the predictions did not rank the active conformation of SLII under the 10% thermodynamically most favorable structures"

- We apologize for not being clear enough at this point. We have rephrased the sentence.

Reviewer #4 (Remarks to the Author):

The HCV 5'UTR contains two binding sites for the liver-specific, highly abundant miR-122. The binding of miR-122 to the 5'UTR enhances stability of the HCV genome, its IRES-dependent translation as well as replication. The molecular mechanism by which all these functions are exerted remained unknown.

In this elegant study, the authors dissect the functions of miR-122 in viral translation and replication.

First they compared two wt HCV 5'UTR luciferase constructs where translation is either HCV IRES-dependent or IRES-independent in Hep3B cells lacking detectable miR-122 expression. Replication in the IRES-dependent construct was only observed upon co-transfection with miR-122 duplex but not in its absence or with a seed mutant of the miRNA. The HCV IRES-independent construct showed efficient replication in a miR-122 independent manner.

Translation, on the other hand, was only increased for the IRES-dependent construct in the presence of miR-

122 duplex but not the seed mutant of the miRNA or the IRES-independent construct. Subtracting the effects of the miRNA on the HCV RNA stability, the authors conclude a 2-3-fold increase in IRES-dependent translation mediated by miR-122. This is further supported with a dicistronic constructs showing that miR-122 functions in stability and efficient translation are linked to efficient HCV replication.

- We are grateful for the very positive perception of our manuscript.

The authors speculate that the miR-122 binding site in the HCV 5'UTR might affect the proper folding of the downstream domain II of the HCV IRES as well as the HCV CRE formed on the negative strand RNA during replication. Comparison of 1D NMR spectra comprising nt 1-120 of the 5'UTR (including the miR-122 binding site) or nt 40-120 (excluding the miR-122 binding site) show increased line broadening for the longer construct indicative of conformational exchange between alternative folds in the presence of domain I.

Comment:

While this is one possible explanation, it is also possible that the RNA simply oligomerizes by forming intermolecular weak base pairing between single-stranded nucleotides in domain I which are stabilized by lower temperatures. This could be easily studied by a simple gel filtration experiment comparing 1-120 and 40-120 constructs at different temperatures. In addition, miR-122 binding should be monitored by NMR. If miR-122 binding really promotes proper folding of domain II by binding the single-stranded region in domain I, the oligomerization should not be observed anymore with RNA 1-120 in the presence of miR-122 and characteristic imino protons of domain II should be clearly visible even at lower temperatures. In case of severe overlap, ¹⁵N labelling would be an option to monitor proper domain II folding induced by miR-122.

- We thank the reviewer for the constructive suggestions. To exclude oligomeric structures we performed size exclusion chromatography combined with online detection by Multi Angle Laser Light Scattering and refractometry. Here we found evidence for monomers and dimers, but not oligomeric structures and miR-122 did bind to the RNA of monomers and dimers, but had no impact on the dimers. We further performed native gelelectrophoresis of the 1-120 fragment incubated at different temperatures, revealing only a single band under all conditions. We have yet not performed NMR studies in presence of miR-122, since all in vitro data (primarily SHAPE) so far indicated that in vitro binding might not be a proper representation of the in vivo situation. However, we are planning such experiments using ¹⁵N labelling as suggested by the reviewer, but this has not been possible within the time frame of the revision of this manuscript. We have included the new data in the revised manuscript (Suppl. Fig. 5) and carefully interpret the results in the text of the revised manuscript.

Comment:

In silico folding analysis suggests that the 1-120 construct forms a different domain II structure which is not translation competent. Unfortunately, SHAPE probing in the presence and absence of miR-122 could not detect significant changes in reactivity in the 1-120 HCV 5'UTR since single-stranded nucleotides are very similar in both predicted folds of the 1-120 construct. Using other probes such as DMS or CMCT might be more useful to confirm the alternative fold since the SLII-alt displays more cytosines, adenines as well as uracils in regions not forming canonical base pairs (see below).

- We are grateful for this valuable suggestion. We have performed experiments with NAI, DMS and NAz. All reagents show a band pattern in agreement with our initial structural prediction, based on NAI and arguing for the initial hypothesis of the existence of an alternative structure in solution. The new data are shown in Suppl. Fig. 4a and 4b.

Nevertheless, the data indicate that the first 40 nt of the 5'UTR are essential for replication but somehow hamper translation in the absence of miR-122. But "currently available biochemical measures" have not been exhausted to provide evidence for an alternative domain II fold.

- This is true. We have rephrased the sentence to indicate that the methods we have used so far do not demonstrate a direct conformational change induced by miR-122.

The authors then design several deletion and point mutations in the miR-122 binding site that should favor proper domain II folding in the absence of miR-122 and thus promote translation. All these mutants fail to replicate but show translational efficiency comparable to wt (in the presence of miR-122) unchanged by miR-122. A mutant lacking the miR-122 binding site and favoring an alternative fold in domain II, on the other hand, translated comparable to wt in the absence of miR-122 and could not be rescued by miR-122 addition. This provides genetic evidence that the proper, miR-122-dependent domain II folding is required for efficient HCV IRES-dependent translation.

P9, line 234:

miR-122 independent translational enhancement facilitates HCV replication.

Dicistronic vector: 1st wt UTR1 needs miR-122 for efficient translation, 2nd wt UTR2 also needs miR-122, but not the other mutants of UTR2 (D20, C, ISL). No replication was observed for any of the mutants without functional miR-122 but in the presence of miR-122 D20 and C mutants restored replication while ISL did not. I do not understand this or I am missing something?

- This is indeed a complicated experiment and we have rephrased this part of the results section to clarify this point. Actually, all constructs in Fig. 5 contain the wt UTR1, binding to miR-122. Therefore, stability of all mutant RNAs (not translation!) still depends on miR-122 binding. Still, miR-122 will also apparently stimulate translation of these RNAs due to increased RNA stability and longer accessibility for ribosomes. In absence of miR-122 we think that the RNA is degraded too fast to initiate replication, although translation by UTR2 is efficient and does not depend on miR-122 for D20 and C. In case of ISL, translation is also not affected by miR-122, but the SLII fold is impaired and therefore translation is generally inefficient, therefore, the replicase proteins are not synthesized (NS3-5B, driven by UTR2) and replication cannot be initiated.

D20 deletes the MBR so how does it work on the negative strand? Where is miR-122 binding in the SLIIz' fold of the negative strand?

- We are not fully clear about this comment. SLIIz' is an essential structure for replication in the negative strand of the genome. It contains the complementary sequence of the MBR, therefore it cannot bind miR-122. We have explained this in more detail in the description and the legend to Fig. 3b and hope that this point is clearer now.

On p8, line 195, D20 was severely impaired in replication; but in the dicistronic vector all replication should occur via wt UTR1 in a miR-122-dependent manner anyway. So how is the difference in replication between D20, C and ISL explained?

- It is true that replication initiates at the first UTR. However, the production of viral protein is dependent on the central IRES element. Consequently, the mutants which stimulate IRES activity produce more NS proteins, which is an essential prerequisite for RNA replication. D20 is not viable in a monocistronic replicon (Suppl. Fig. 7b) since it deletes essential replication elements of the negative strand (part of SLIIz', Fig.3b). The positioning of the D20 variant at UTR2 allows us now to assess the contribution of the translation efficiency of the IRES mutants to the overall replication efficiency. ISL favors SLIIalt formation and shows decreased translational activity, whereas C and D20 favor formation of SLII and stimulate translation, in all cases independent of miR122 (Fig.4). The fact that a Bi-Luc-SG replicon with ISL at position of UTR2 is not replication competent, whereas D20 and C are, provides further support for substantial contribution of DI to IRES activity (all mutations are outside the IRES in DI, Fig. 4). More importantly, mutations D20 and C can replace miR-122 function on translation in UTR2 and this further demonstrates that miR-122 mediated stimulation of translation is essential for replication (since ISL at this position is not replication competent). We have carefully rephrased this section to make this concept clearer for the reader.

P9, line 249: What is the evidence that translation stimulation by miR-122 contributes to HCV replication? For me these are independent processes, on the positive strand, miR-122 promotes and active IRES, on the negative strand, replication through interaction with a different RNA, where is the link?

- As already explained in response to the previous point this was the central idea underlying the BiLuc-SG constructs. Since they contain two copies of the UTR, we can address functions in replication (UTR1) and translation (UTR2) separately (see also Fig. 2 and associated text). For initiation of replication efficient translation of viral proteins must occur first. Efficient translation at UTR2 requires miR-122 (Fig. 2) or mutants favoring the structure of SLII (D20 and C, but not ISL, Fig. 5). Therefore, D20 and C at position of UTR2 render BiLuc-SG constructs replication competent, ISL in contrast not. Overall, the data shown in Fig. 2 and 5 therefore support our assumption, that stimulation of translation (by stabilizing SLII in UTR2) is an important contribution of miR-122 to RNA replication, in addition of stabilizing the genome by preventing degradation by Xrn1 (UTR1). As mentioned above, all functions of miR-122 are due to binding to the MBR in the 5'UTR of the positive strand genome, miR-122 has no binding site on the negative strand. We have revised the text of this section to make this important point clearer (see above).

P3, line 50: Should be: which positions the viral AUG start codon in the ribosomal P site.

P9, line 250: Should be: miR-122

- This has been corrected.

Reviewers' Comments:

Reviewer #1 (Remarks to the Author):

I am happy with the authors' response to my original comments.

Reviewer #2 (Remarks to the Author):

The authors have carefully addressed previously raised concerns. This is a well-executed study.

Reviewer #3 (Remarks to the Author):

The revision has improved the manuscript, which now includes a more clear mentioning of the caveats associated with the study. I find that my major concern has been taken care of and recommend that the paper is accepted for publication.

Reviewer #4 (Remarks to the Author):

The authors have addressed all questions and comments raised.